# Structural insights into tecovirimat antiviral activity and poxvirus resistance

Riccardo Vernuccio [1], Alejandro Martínez León [2], Chetan S. Poojari [2], Julian Buchrieser [3], Christopher N. Selverian [4], Yakin Jaleta[4], Annalisa Meola[1], Florence Guivel-Benhassine[3], Françoise Porrot [3], Ahmed Haouz [5], Maelenn Chevreuil[6], Bertrand Raynal [6], Jason Mercer[7], Etienne Simon-Loriere [8], Kartik Chandran [4], Olivier Schwartz [3], Jochen S. Hub [2] & Pablo Guardado-Calvo [1] ✉

Mpox is a zoonotic disease endemic to Central and West Africa. Since 2022, two human-adapted monkeypox virus (MPXV) strains have caused large outbreaks outside these regions. Tecovirimat is the most widely used drug to treat mpox. It blocks viral egress by targeting the viral phospholipase F13; however, the structural details are unknown, and mutations in the F13 gene can result in resistance against tecovirimat, raising public health concerns. Here we report the structure of an F13 homodimer using X-ray crystallography, both alone (2.1 Å) and in complex with tecovirimat (2.6 Å). Combined with molecular dynamics simulations and dimerization assays, we show that tecovirimat acts as a molecular glue that promotes dimerization of the phospholipase. Tecovirimat resistance mutations identified in clinical MPXV isolates map to the F13 dimer interface and prevent drug-induced dimerization in solution and in cells. These findings explain how tecovirimat works, allow for better monitoring of resistant MPXV strains and pave the way for developing more potent and resilient therapeutics.

Orthopoxviruses (OPXVs) produce several human diseases, including smallpox and mpox, caused by variola (VARV) and monkeypox (MPXV) virus, respectively. Cowpox (CPXV), camelpox (CMLP), and borealpox (BRPV) also lead to sporadic zoonotic infections[1]. Smallpox vaccination was discontinued 50 years ago, leaving most of the current population unprotected and raising concerns about the emergence of zoonotic OPXVs or the reintroduction of VARV. Two major mpox epidemics have emerged since 2022. The first, caused by a mild clade II strain, spread rapidly across the globe, resulting in more than 90,000 cases and 179 deaths[2]. The second, produced by a virulent clade I strain, produced the largest outbreak ever recorded in the Democratic Republic of Congo and spread to neighbouring countries, resulting in hundreds of deaths[3]. First-generation vaccines do not meet modern safety standards, and attenuated third-generation vaccines, although effective, are difficult to produce on a large scale and do not induce long-term immunity[4,5]. Most recently, mRNA-based candidate vaccines have shown efficacy in animal models and are currently being tested in clinical trials[6,7].

OPXVs have an unusual replication cycle that produces two different virions termed mature and enveloped viruses. Mature viruses are produced in the cytoplasm and formed by a viral capsid surrounded

[1]G5 Structural Biology of Infectious Diseases, Institut Pasteur, Université Paris Cité, Paris, France. [2]Theoretical Physics and Center for Biophysics, Saarland University, Saarbrücken, Germany. [3]Virus and Immunity Unit, Institut Pasteur, Université Paris Cité, CNRS, UMR 3569, Paris, France. [4]Department of Microbiology and Immunology, Albert Einstein College of Medicine, Bronx, NY, USA. [5]Crystallography Platform-C2RT, UMR 3528, Institut Pasteur, CNRS, Université de Paris, Paris, France. [6]Plate-forme de Biophysique Moleculaire-C2RT, Institut Pasteur, CNRS UMR 3528, Université Paris Cité, Paris, France. [7]Institute of Microbiology and Infection, School of Biosciences, University of Birmingham, Birmingham, UK. [8]G5 Evolutionary Genomics of RNA Viruses, Institut Pasteur, Université Paris Cité, Paris, France. ✉e-mail: guardado@pasteur.fr

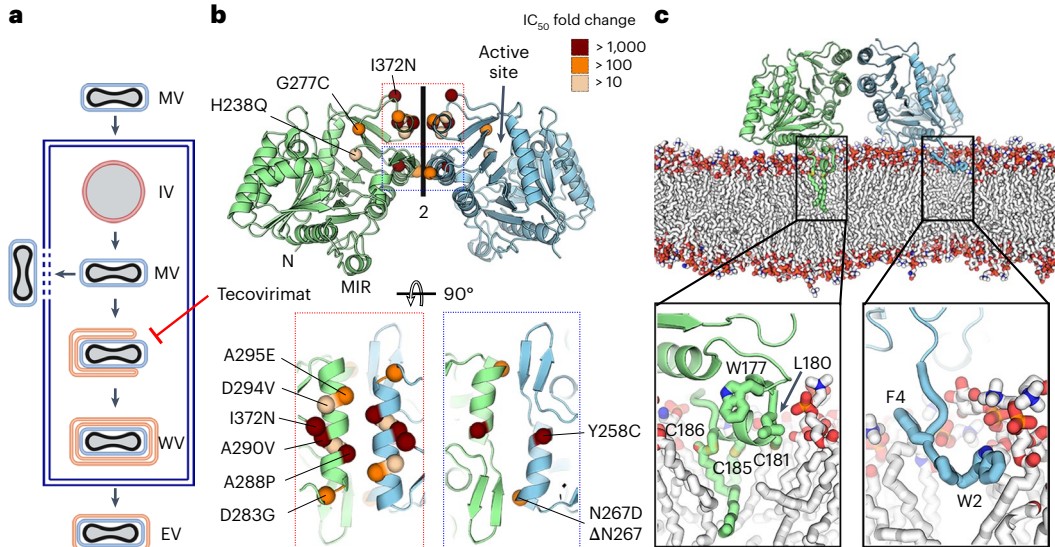

**Fig. 1 | F13 forms a homodimer that can be inserted into a membrane's surface.** **a**, Schematic representation of the replication cycle of OPXVs. Mature viruses enter the cell, fusing their membrane (in blue) with the cellular one. After DNA replication, immature particles are formed (IV, membrane in red), which give rise to intracellular mature virus particles in the cytoplasm of the infected cell. Mature viruses (MV) can either be released by lysis or wrapping. In the latter, mature viruses acquire two additional membranes (WV, in orange) from the Golgi apparatus or endosomal vesicles to form wrapped virions, fuse the outermost with the plasma membrane and release enveloped viruses (EV). Tecovirimat blocks wrapping, as indicated. The scheme, adapted from Fig. 1 in ref. 86, was created with BioRender.com. **b**, Crystal structure of the sF13 homodimer represented in cartoon. One protomer is coloured blue and the other green. The N termini and the MIR are indicated on one protomer, and the phospholipase active site is indicated on the other. Bottom panels provide close-up views of the two regions forming the dimer interface, indicated by coloured rectangles in the upper panel. All single escape mutants identified to date are shown as spheres, coloured according to their potency, reported as IC50 fold change. **c**, Side view of the F13 homodimer interacting with a lipid membrane that mimics Golgi membrane composition, as observed from molecular dynamics simulations. For clarity, water molecules and lipids in the foreground of the membrane are not shown. sF13 chains are coloured as in **b**, with palmitoylated cysteines and hydrophobic residues in the MIR and N termini depicted as sticks. The bottom panel provides close-up views to show lipid–protein interactions, with the protein residues involved in the interaction depicted as sticks and labelled. Protein carbons are coloured according to the chain, membrane carbons in white. Nitrogen, oxygen, sulfur and phosphate atoms are coloured blue, red, yellow and orange, respectively.

by a single membrane. To disseminate within the host, mature viruses form wrapped virions covered by three membranes, which henceforth fuse the outermost with the plasma membrane to leave the host cell as enveloped viruses, covered by two remaining membranes (Fig. 1a). Two oral drugs have been approved for the treatment of smallpox and mpox, brincidofovir (Tembexa) and tecovirimat (TPOXX). Brincidofovir is an inhibitor of viral DNA polymerase that has been shown to produce side effects in patients[8]. Tecovirimat is an inhibitor of mature virus wrapping widely used to treat mpox patients infected with clade IIb strains. However, it has a low resistance barrier, and multiple tecovirimat-resistant MPXV strains have been reported[9,10]. Resistance mapping studies[10] have indicated that tecovirimat targets the virus envelope protein F13, a membrane-anchored phospholipase that plays a key role in the production of wrapped virions[11,12]. Tecovirimat binding to F13 blocks viral wrapping[13], but the structural mechanism is poorly understood. Molecular dynamics simulations based on predicted structures have suggested different tecovirimat binding sites[14,15], but none of them explain either how tecovirimat blocks wrapping or why resistance mutants escape the drug. This paucity of structural and mechanistic data prevents anticipating which mutations may confer resistance to tecovirimat and the development of better drugs.

In this article, we elucidate using crystallography and molecular dynamics simulations the structure of F13 with tecovirimat. We showed that F13 forms a homodimer on membranes, and tecovirimat inserts into a cavity formed between the two protomers. Using analytical ultra-centrifugation (AUC), size-exclusion chromatography coupled with small-angle X-ray scattering (SEC-SAXS), mass photometry, proximity ligation assay (PLA) and binding free energy calculations, we further report that tecovirimat induces homodimerization of F13 in solution

and within cells, but not in resistant mutants, providing a mechanistic basis for drug activity and viral escape.

## Results

### Tecovirimat binds a pocket between two protomers

F13 is a phospholipase anchored to membranes via two palmitoylated cysteines located in a hydrophobic membrane-interacting region (MIR; Extended Data Fig. 1)[16]. For the structural studies, we produced a soluble variant of F13 (sF13) by removing the hydrophobic N-terminal tail (amino acids 2–5) and by introducing five mutations in the MIR. We obtained two different crystal forms (Supplementary Table 1). In both, sF13 featured a homodimer stabilized by two helices and a β-hairpin (Fig. 1b) with an interface area of 939 Å², formed by a network of hydrogen bonds involving residues Y253, N259, N267, Y285, S292 and N300 (Extended Data Fig. 2). The contact region forms a large cavity of 290 Å³. The hydrophobic N termini and MIRs were located on one side of the dimer, and the two phospholipase D (PLD) catalytic pockets were pointed outwards (Fig. 1b). To evaluate whether the dimer was compatible with membrane insertion, we performed molecular dynamics simulations on membranes mimicking the composition of the Golgi membrane (Supplementary Table 2). We observed that the homodimer remained intact during the 1,000 ns of simulation, suggesting that the dimeric interface is stable in the physiological membrane-bound state. sF13 was anchored to the membrane with the two palmitoylated cysteines inserted deeply into the outer leaflet of the membrane, while the N-terminal tail was associated with the lipid head group and glycerol regions (Fig. 1c). Two lines of evidence suggest that the homodimer has a biological role: (a) other membrane-interacting PLDs, such as the human exonucleases PLD3 and PLD4 (refs. 17,18), form

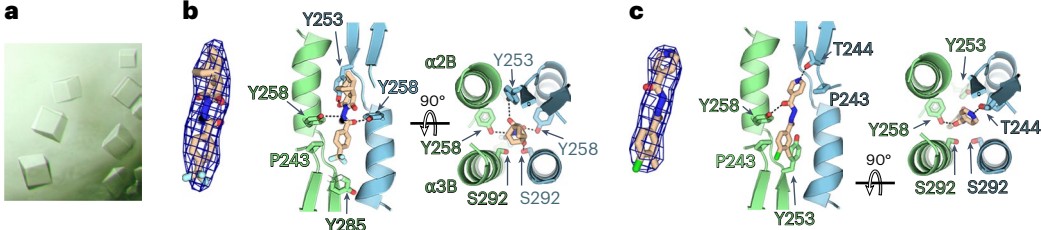

**Fig. 2 | Tecovirimat binding site. a**, Cubic crystals used to obtain the structure of the sF13/tecovirimat and sF13/IMCBH complexes. **b**, Crystal structure of the sF13/tecovirimat complex. Left: a Fo-Fc omit map contoured at 3σ showing the density found at the dimer interface in the soaked crystal with the tecovirimat molecule modelled. Centre and right: orthogonal views of the dimerization interface with the tecovirimat molecule modelled and the residues contacting the drug represented as sticks and labelled. sF13 chains are coloured as in Fig. 1. **c**, Crystal structure of the sF13/IMCBH complex. Left: as in **b**, an omit map showing the electron density at the dimer interface in the soaked crystal. Centre and right: as in **b**, orthogonal views showing the sF13/IMCBH contacts.

similar homodimers with physiological roles; and (b) mapping single escape mutants onto the sF13 homodimer reveals that all lie within or close to the dimer interface. Four escape mutations known to produce higher resistance to tecovirimat (Y258C, A288P, A290V and I372N) were located at the centre of the interface (Fig. 1b and Supplementary Table 3). Moreover, an escape mutant (D280Y) of $N_1$-isonicotinoly-$N_2$-3-methyl-4-chlorobenzoylhydrazine (IMCBH), a selective inhibitor of vaccinia virus (VACV) replication that blocks wrapped virion formation in vitro[19,20], also mapped close to the dimer interface. These data indicate that F13 dimerizes when present on membranes and that this homodimer is targeted by tecovirimat and IMCBH.

To map the tecovirimat binding site, we soaked cubic crystals (Fig. 2a) in a solution containing 1 mM tecovirimat. The resulting maps revealed an additional electron density at the dimeric interface compatible with the size and shape of a tecovirimat molecule (Fig. 2b). However, because the pocket is symmetrical and tecovirimat is an asymmetric molecule, the resulting electron density is featureless, preventing accurate modelling of the molecule. To identify the correct pose and rationalize the featureless electron density, we performed absolute binding affinity calculations based on molecular dynamics simulations and free energy perturbation (FEP) techniques. We generated 15 poses of different rotational states of tecovirimat compatible with the electron density. After equilibration of each pose using free molecular dynamics simulations, we computed the absolute binding free energy (ABFE) using three independent sets of full FEP calculations. Because FEP achieves ABFEs with an accuracy of approximately 1–2 kcal mol⁻¹ (ref. 21), the calculations enabled us to identify high-affinity poses selected by tecovirimat in the experiment. It is worth noting that we identified multiple poses with very low binding free energy values ($\Delta G_{bind}$) of less than −20 kcal mol⁻¹ (Extended Data Fig. 3 and Supplementary Table 4). Averaging over 45 independent simulations, we estimated an exceptionally strong affinity of $\Delta G_{bind}$ = −25.1 kcal mol⁻¹, suggesting that tecovirimat strongly stabilizes the homodimer. The presence of multiple conformers with similarly high affinities is compatible with the presence of multiple tecovirimat conformers in the crystal, in agreement with the featureless electron density. To validate our approach, we evaluated a set of seven structurally similar ligands with published half-maximal effective concentration (EC50) values alongside tecovirimat[22,23]. We aligned each ligand to the tecovirimat best pose and performed ABFE calculations. We obtained good agreement between the computed ABFE values and experimental data (Extended Data Fig. 4).

We used the highest affinity pose to refine the crystallographic data. We observed that the molecule was stabilized by a network of polar contacts involving Y258 and S292 and hydrophobic contacts mediated by Y253, I262, I266 and Y285 (Fig. 2b). Similarly, we soaked sF13 cubic crystals with IMCBH and observed an electron density between both protomers that was compatible with the dimensions of IMCBH (Fig. 2c). Overall, the structural data revealed that both

tecovirimat and IMCBH bind to the same pocket between the two protomers of the homodimer. The estimated $\Delta G_{bind}$ values suggest that tecovirimat stabilizes F13 homodimers, like a molecular glue.

## Tecovirimat stabilizes F13 homodimers

Next, we evaluated the oligomeric state of sF13 with and without tecovirimat by using AUC. In the absence of the drug (Fig. 3a), sF13 was predominantly monomeric in solution (sedimentation coefficient, $S$ = 3.4), with a minor presence of a higher $S$ coefficient species ($S$ = 4.4), which we interpreted as a dimer. Consistent with our hypothesis, tecovirimat shifted the equilibrium to the dimeric form ($S$ = 4.7), with no monomeric protein detected. To confirm that the tecovirimat-induced dimer in solution corresponded to the dimer observed in the crystals, we analysed the sF13/tecovirimat complex using SEC-SAXS (Fig. 3b and Extended Data Fig. 5). In agreement with the AUC experiments, the F13/tecovirimat complex behaved as a monodisperse distribution of dimers in solution with an estimated molecular mass of approximately 76–84 kDa, a maximum distance ($D_{max}$) of 108 Å and a radius of gyration ($R_g$) of 32.9 Å, matching the calculated values of the crystallographic dimer of 88.6 kDa, 117 Å and 32.3 Å, respectively. For further structural insight, we compared the experimental SAXS curve with the theoretical curves of the F13 monomer and dimer (Fig. 3b) and the ab initio model derived from the SAXS curve with the crystallographic F13 dimer (Fig. 3c). We found in both analyses that the crystallographic dimer fitted well to the experimental SAXS curve. Overall, these results show that tecovirimat induces the dimerization of sF13 in solution and that the resulting dimer adopts the same organization as the dimer observed in the crystal structure.

Next, we assessed the activity of tecovirimat in solution. To do this, we designed an assay to calculate the concentration of the drug required to induce dimerization of 50% of sF13 (EC50). We used mass photometry, which measures the molecular mass of individual molecules by quantifying the light scattering in a dilute solution. This technique enables a higher throughput than AUC or SEC-SAXS and allows for the evaluation of a full range of tecovirimat concentrations. We compared the EC50 values obtained using the mass photometry assay: 92 nM for tecovirimat and 1,475 nM for IMCBH (Fig. 3d and Extended Data Fig. 6) to the half-maximal inhibitory concentration (IC50) values measured in cells infected with a recently isolated clade IIb MPXV strain[24]: 17 nM for tecovirimat and 74 nM for IMCBH (Fig. 3e). In both assays, tecovirimat outperformed IMCBH, indicating that the activity measured in solution correlated with the antiviral activity of the drugs. The higher EC50 values in solution can be explained by higher entropy loss during dimerization as, on the membrane surface, F13 diffused in only two translational dimensions and one rotational dimension. Tecovirimat is approximately 15 times more active in solution than IMCBH, but its antiviral activity is 5 times better. This discrepancy could be explained by factors unrelated to their biochemical activities, such as differences in membrane permeability or cellular distribution.

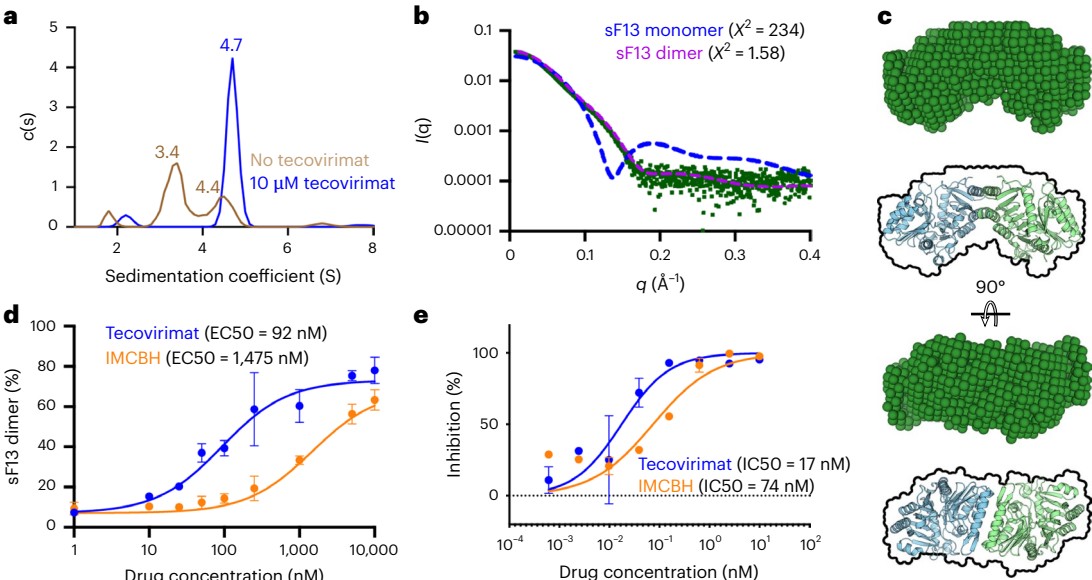

**Fig. 3 | Tecovirimat induces sF13 dimerization in solution. a**, AUC analysis of sF13 without tecovirimat (brown line) and with 10 μM tecovirimat (blue line). Experimentally derived sedimentation coefficient values (Svedberg units (S)) are shown above each peak. $c(s)$, the concentration of protein with sedimentation coefficient s. **b**, Experimental SAXS profile (green dots) and theoretical profiles (dashed lines) calculated using CRYSOL[87] for one monomer of sF13 (blue dashed line) and the dimer shown in Fig. 1 (pink dashed line). $I(q)$, scattering intensity in function of the scattering vector q; $q$ (Å$^{-1}$), scattering vector; Å, Angstrom; $X^2$, chi-squared test. **c**, Orthogonal views of a representative dummy atom model (green) reconstructed from SAXS data. For comparison, we have included below a model of the crystallographic sF13 dimer with an outline of the dummy model.

**d**, Dose–response curve used to estimate tecovirimat effect in solution. The $y$ axis represents the proportion of dimers in a dilute solution of F13 measured by mass photometry. The $x$ axis represents the concentration of drug (tecovirimat or IMCBH) present in the solution. The EC50 values were determined from a dose–response curve fitted using GraphPad Prism. Data are mean ± s.d. of three independent experiments ($n = 3$). **e**, Tecovirimat (blue line) and IMCBH (orange) inhibit plaque formation of MPXV. Vero cells were infected with MPXV clade IIb and treated with the indicated concentrations of tecovirimat or IMCBH. Plaque inhibition is expressed as a percentage, normalized to control conditions. Data are mean ± s.d. of triplicate wells from five independent experiments for tecovirimat ($n = 15$) and two independent experiments for IMCBH ($n = 6$).

To identify the pathway by which tecovirimat induces dimerization in solution, whether it binds to the monomer and induces dimerization or binds to a pre-formed dimer, we computed the binding affinity of tecovirimat to F13 protomers. Using the best tecovirimat pose, we removed one protomer from the molecular dynamics simulation system and computed the affinity for the remaining protomer. We estimated a $\Delta G_{bind}$ of −5.7 kcal mol$^{-1}$ (Supplementary Table 5), which corresponds to an EC50 value of approximately 68 μM. Thus, at the EC50 value of dimerization of 92 nM (Fig. 3d), only a small fraction of F13 protomers will be occupied by tecovirimat, suggesting that tecovirimat binds and stabilizes a transient dimer.

**Tecovirimat escape mutants prevent F13 dimerization**

Taken together, these structural and functional data suggest that escape mutants circumvent drug activity by preventing the formation of F13 homodimers. To test this hypothesis, we investigated the effect of escape mutants on tecovirimat activity using the mass photometry assay. We tested three escape mutants identified in mpox patients treated with tecovirimat (Supplementary Table 3): A295E, the quadruple mutant N267D, A288P, A290V, D294V (4MUT) and ΔN267. We also assayed one escape mutant (G277C) identified in vitro but never reported in mpox patients. The mutations identified in the clinical MPXV strains were located at the homodimerization interface and were expected to have an impact on tecovirimat-induced dimer formation, whereas G277C was located further away (Fig. 1b). Consistently, the mass photometry assay showed that sF13$^{A295E}$, sF13$^{4MUT}$ and sF13$^{\Delta N267}$ did not form homodimers in the presence of tecovirimat at the range of concentrations tested, whereas sF13$^{G277C}$ did as the wild-type protein (Fig. 4a). To complement the mass photometry data, we repeated the AUC experiments reported above using sF13$^{A295E}$ and sF13$^{4MUT}$ (Fig. 4b and Extended Data Fig. 6). In contrast to the mass

photometry experiments, which were performed using a dilute solution of sF13 (25 nM), the AUC experiments were performed at a higher concentration (10 μM), to detect weaker tecovirimat activities. Thus, in the AUC experiments, we observed that the drug induced partial dimerization of sF13$^{A295E}$, but not of sF13$^{4MUT}$, which aligns with the ΔEC50 values reported for A295E and 4MUT.

Next, we aimed to elucidate the structural basis of the partial resistance of the A295E mutant. We obtained cubic crystals of sF13$^{A295E}$ and soaked them in tecovirimat. In the absence of tecovirimat, sF13$^{A295E}$ formed a homodimer that resembled that of sF13$^{WT}$, but with the ends of the α10 helix being more open, so that Y285 was hydrogen-bonded with Q299 instead of with N300, as in the wild-type protein (Fig. 4c). This results in a reduction in the buried surface area from 939 to 882 Å$^2$, likely reducing dimer stability. In the presence of tecovirimat, sF13$^{A295E}$ recovered the native conformation, in which Y285 was hydrogen-bonded to N300 (Fig. 4d). We failed to crystallize sF13$^{4MUT}$, but it is likely that the introduction of a proline in the middle of the α10 helix generates a conformational reorganization that prevents F13 dimerization. Overall, we concluded that escape mutants isolated from tecovirimat-treated patients alter the dimerization interface of sF13, making tecovirimat-induced dimerization less efficient.

We then investigated whether tecovirimat induces F13 dimerization in cells. To evaluate this, we performed PLA. This technology is based on two oligonucleotide-labelled antibodies (probes) that bind to the constant region of a pair of primary antibodies targeting the proteins of interest. If the probes are less than 40 nm apart, they hybridize and produce a fluorescent signal that can be visualized and quantified by fluorescence microscopy (Fig. 5a). As we did not have specific antibodies targeting F13, we engineered a version of F13 with a flag tag in the loop connecting β1A to α1A, away from the dimerization

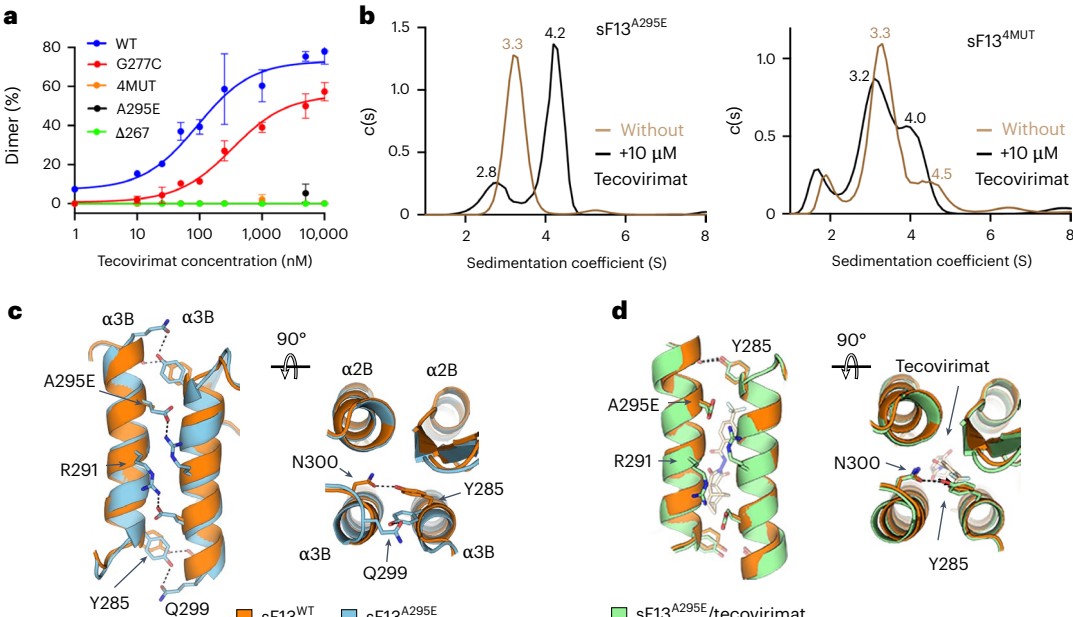

**Fig. 4 | Escape mutants identified in mpox patients prevent tecovirimat-induced dimerization. a**, Mass-photometry-based dose–response curve showing tecovirimat activity against different escape mutants, as indicated. Data are mean ± s.d. of three independent experiments ($n = 3$). **b**, AUC analysis of sF13$^{A295E}$ (left panel) and sF13$^{4MUT}$ (right panel) without tecovirimat (brown line) and with tecovirimat (black line). Experimentally derived

sedimentation coefficient (S) values are shown above each peak. **c,d**, Orthogonal views showing the dimer interface of sF13$^{A295E}$ (cyan, **c**) and sF13$^{A295E}$/tecovirimat (green, **d**) superimposed on sF13$^{WT}$ (orange). Residues E295, R291, Y285 and N300 are represented as sticks and labelled. Polar contacts are indicated with dashed lines.

and membrane interaction interfaces (Flag–F13$^{WT}$). We also produced two variants that escape tecovirimat (Flag–F13$^{4MUT}$ and Flag–F13$^{A295E}$). We used two commercial antibodies targeting the flag tag, one with a mouse and the other with a rabbit Fc. Hela cells were transiently transfected with Flag–F13$^{WT}$, Flag–F13$^{4MUT}$ or Flag–F13$^{A295E}$ and incubated with 10 μM tecovirimat or 0.1% dimethyl sulfoxide (DMSO) as a control. PLA signal was quantified after 24 h of incubation. A signal was detected in F13-expressing cells but not in control cells. In the absence of tecovirimat, a signal was observed in cells expressing WT or mutant F13, likely corresponding to basal levels of proteins that were less than 40 nm apart. In line with the AUC experiments, tecovirimat induced a strong increase in the PLA signal in cells expressing Flag–F13$^{WT}$, a slight increase in cells expressing Flag–F13$^{A295E}$ and none in cells expressing Flag–F13$^{4MUT}$ (Fig. 5b and Extended Data Fig. 7). Therefore, we conclude that tecovirimat induces F13 dimerization in cells and that escape mutants interfere with this dimerization.

**Dimer-stabilizing mutations render VACV non-viable**

Isolation of tecovirimat-resistant MPXV strains revealed only ten mutations that, either isolated or in combination, confer resistance to the drug (Fig. 1a and Supplementary Table 3)[9]. We wondered why other mutations did not emerge. One plausible explanation is that most mutations in this region are associated with a major loss of fitness, rendering the virus non-viable. To test this, we designed three escape mutants (S292F, S292K and L296Y) by introducing bulky side chains into the tecovirimat binding cavity (Fig. 6a) and studied their sensitivity to tecovirimat in vitro using the mass photometry assay. We confirmed that sF13$^{S292F}$, sF13$^{S292K}$ and sF13$^{L296Y}$ were totally insensitive to tecovirimat activity (Fig. 6b). sF13$^{S292F}$ and to lesser extent sF13$^{L296Y}$ formed dimers in the absence of the drug, suggesting that filling the cavity with hydrophobic side chains stabilizes the dimer. To assess viral viability, we introduced S292F into VACV using the marker-free vaccinia virus engineering (MAVERICC) system[25] to produce the recombinant virus rVACV$^{S292F}$. As controls, we generated three additional viruses bearing

substitutions in F13 known to confer tecovirimat resistance: rVACV$^{G277C}$, rVACV$^{4MUT}$ and rVACV$^{A295E}$. As expected, all the control viruses could be rescued and grew to high titres (Fig. 6c). However, despite multiple attempts, we were unable to generate rVACV$^{S292F}$, suggesting that this substitution is deleterious to viral morphogenesis.

**Identification of previously undescribed potential tecovirimat-escape mutants**

We then sought for other tecovirimat resistance mutations that might have been unnoticed. To identify them, we extracted the F13 sequence from all MPXV genomes available in the GISAID (Global Initiative on Sharing All Influenza Data) database[26] and from all OPXV sequences available in the GenBank database. Within each species, we mapped variations in the F13 amino acid sequences at the dimer interface. No escape mutants were identified in MPXV clade I sequences. However, we identified three potential escape mutants in MPXV clade II sequences: L296F, D280Y and P243A, all from an immunocompromised patient treated with tecovirimat[27]. In addition, we identified R291E in one out of 81 analysed VARV strains. A structural analysis showed that R291E introduced two negatively charged residues facing each other on both sides of the interface (Fig. 6a), creating electrostatic repulsion that may hinder dimer formation and confer tecovirimat resistance. To test this, we produced sF13$^{R291E}$ and studied its sensitivity in vitro to tecovirimat. We showed that the drug had some activity at the highest concentrations in the assay (Fig. 6b). Next, we evaluated the resistance of rVACV$^{R291E}$ to 10 μM tecovirimat by plaque assay (Fig. 6c) using rVACV$^{WT}$, rVACV$^{4MUT}$ and rVACV$^{A295E}$ as controls. As reported previously[28], tecovirimat abrogated plaque formation by rVACV$^{WT}$. The control mutants completely (A295E, 4MUT) or partially (G277C) escaped tecovirimat, as determined by measurements of both plaque number and plaque area (Fig. 6c), providing evidence that F13 substitutions conferring tecovirimat resistance in MPXV are also transferable to VACV. The mutant R291E remained sensitive to tecovirimat, at least at the high drug concentration we used in these assays.

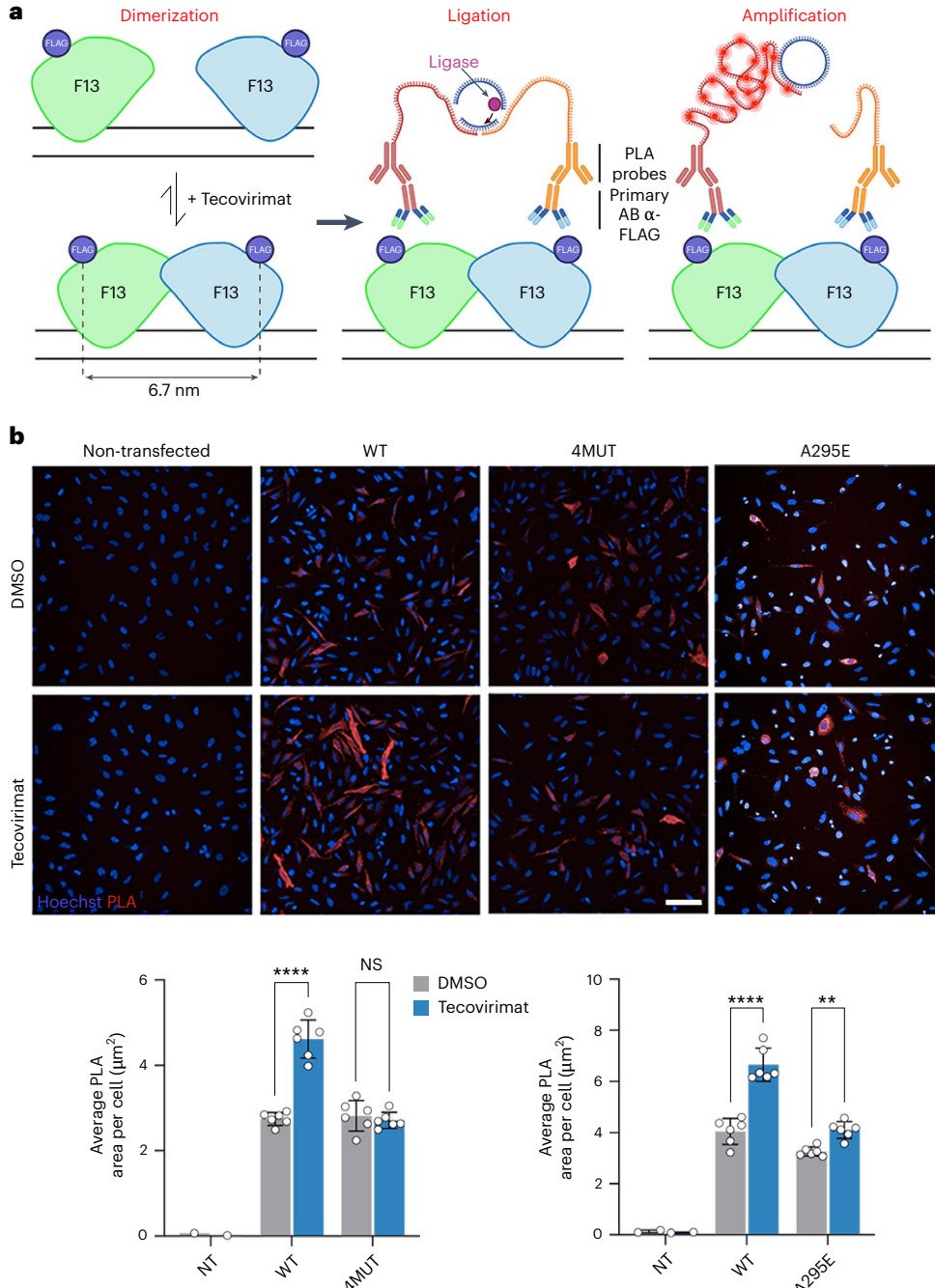

**Fig. 5 | Tecovirimat induces F13 dimerization in cells. a**, Schematic model representing the PLA experiment. F13 protomers are coloured green and cyan with the approximate location of the flag tag indicated with a blue sphere. The three steps of the assay, dimerization, ligation and amplification, are indicated. **b**, Upper panel: representative fluorescence microscopy images with the nuclei coloured in blue and the PLA signal in red. Scale bar, 100 μm. Lower panel: quantification of the PLA signal as the average area of PLA fluorescence per cell; 7,000 to 12,000 cells were analysed per data point. Data are mean ± s.d. of two independent experiments performed in triplicate ($n = 6$). For statistical analysis, two-way analysis of variance was used. NS, non-significant; ****$P < 0.0001$; **$P = 0.0087$. NT, non-transfected.

## Discussion

Here we present a structural study of viral phospholipase F13, including its interaction with tecovirimat. Previous studies have identified three important regions for F13 activity: two palmitoylated cysteine residues required for membrane association[16], a phospholipase motif[29–31] and a di-aromatic motif required for interaction with late endosome proteins[11]. Similar to tecovirimat binding, mutations in any of these regions reduce wrapped virion formation[11,32], but none of the escape mutants identified map there. Our structural

and molecular dynamics data suggest that F13 may homodimerize on membranes. Comparison with other phospholipases revealed that this homodimer closely resembles that of phospholipases PLD3 and PLD4[17,18], which are transmembrane proteins with exonuclease activity. Sequence analysis showed that poxviral phospholipase K4, a paralogue of F13 with nuclease activity[33], shares 48% sequence identity with human PLD3. Taken together, these findings suggest that poxviruses captured a PLD3-like gene from which both viral phospholipases evolved; K4 became a soluble protein but maintained

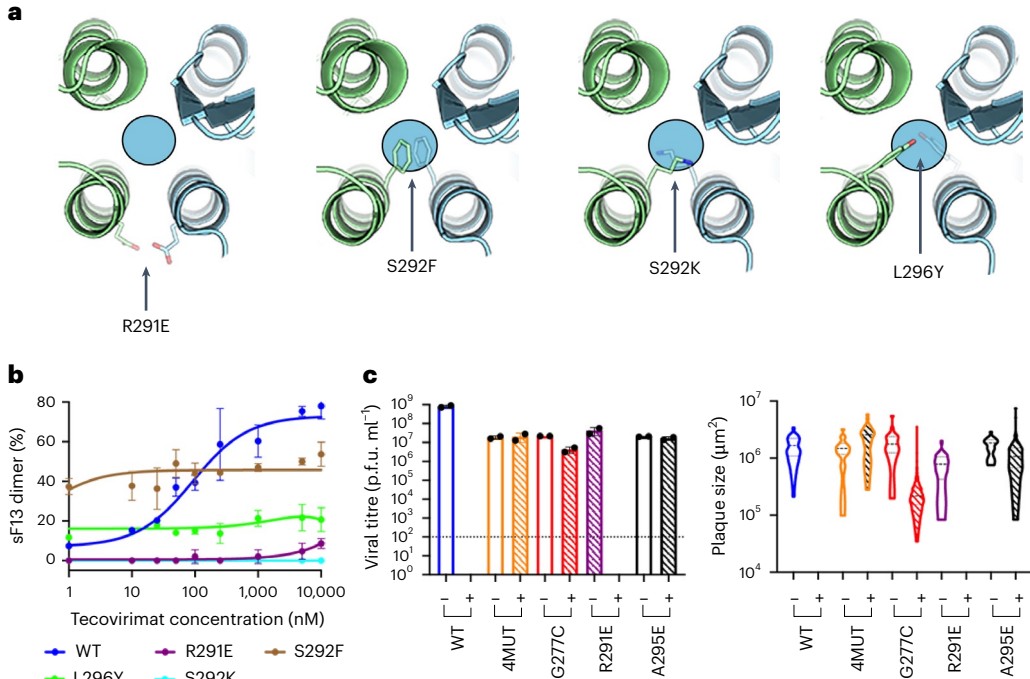

**Fig. 6 | Structure-based escape mutations do not generate viable viruses.**
**a**, Close view of the dimerization interface across the twofold axis showing the designed mutations S292F, S292K and L296Y and the mutation identified in VARV, R291E. The circle indicates the localization of the tecovirimat-binding site. **b**, Mass-photometry-based dose–response curve showing tecovirimat activity against different mutants, as indicated. Data are mean ± s.d. of three independent experiments ($n = 3$). **c**, Viral titres in p.f.u. ml$^{-1}$ (left panel) and plaque size (right panel) in the presence (+) and absence (−) of 10 μM tecovirimat calculated from plaque assays. Each bar represents the means ± s.d. from two independent experiments ($n = 2$). The limit of detection ($10^2$ p.f.u. ml$^{-1}$) is indicated with a dashed line.

its nuclease activity, whereas F13 remained in the membrane but acquired broad phospholipase activity[31].

Despite this evolutionary link, there is an important difference between the homodimer of F13 and its eukaryotic counterparts. The active form of PLD3 is a stable dimer in solution[17], but the soluble version of F13 is mostly monomeric. Structural analysis revealed a large cavity at F13's dimerization interface, which is absent in PLD3 and PLD4 homodimers. We hypothesized that this cavity reduces the stability of the F13 dimer. Sequence analysis showed that this cavity is conserved across OPXVs, and most of the tecovirimat escape mutants were located around it. Molecular dynamics simulations showed that tecovirimat binds to the cavity with strong affinity, with some poses exhibiting binding free energies lower than −20 kcal mol$^{-1}$, demonstrating that tecovirimat serves as a highly potent molecular glue. AUC, SEC-SAXS, mass photometry and PLAs confirmed that tecovirimat induced dimerization of F13, both in solution and within cells.

This model provides a molecular framework for understanding most escape mutants identified thus far; they alter the F13 dimerization region and prevent tecovirimat from inducing dimerization. To support this hypothesis, we developed a mass photometry assay to measure the activity of tecovirimat in solution and showed that A295E, 4MUT and ΔN267 did not dimerize in the presence of the drug.

The results presented here have broad implications for public health. Estimating tecovirimat sensitivity of clinical isolates is important for monitoring epidemics. This is currently performed by isolating the virus[24,34], which is labour intensive and requires expensive and advanced equipment. The precise mapping of F13/tecovirimat contacts provides the possibility of performing sequence-based estimations of tecovirimat sensitivity. While we were preparing this paper, early results from the PALM007 clinical trial were released[35], showing that tecovirimat did not accelerate recovery in patients infected with MPXV clade I. Based on the structure of the sF13/tecovirimat complex, we did

not identify any mutations in clade I strains that could explain this lack of activity. Additional clinical trials will examine tecovirimat's effectiveness on different strains and when administered early in infection. However, these results emphasize the need for new antiviral drugs. The structural data, molecular dynamics simulations and the battery of assays developed here will pave the way for the development of these novel antivirals active against tecovirimat-resistant strains.

## Methods

### Protein production and purification

To produce soluble F13 (sF13$^{WT}$ and mutants), we cloned a synthetic gene, codon-optimized for expression in *Escherichia coli* into a pET-28a(+) vector (Novagen) with an N-terminal His- and Strep-tag followed by a thrombin site. The sequence is derived from the Western Reserve strain of VACV (Uniprot code, P04021). We removed the hydrophobic N-terminal tail (residues 2–5) and introduced five mutations in the MIR: W177A, L178A, C181A, C185A and C186A, to remove the palmitoylation sites and the hydrophobic residues around. The point mutants mentioned in the text were introduced into this vector. We transformed *E. coli* BL21 (DE3) cells (New England Biolabs) and induced protein expression overnight at 16 °C with 0.25 mM isopropyl β-D-1-thiogalactopyranoside. We collected cells from 3 l of culture, resuspended them in 40 ml cold resuspension buffer (Tris–HCl 10 mM pH 8, NaCl 150 mM, EDTA 1 mM) supplemented by one tablet of complete protease inhibitor (Pierce) and froze them at −20 °C. The next day, we thawed and lysed them using a sonicator. After removing the insoluble material by centrifugating at 20,000 g (30 min, 4 °C), we purified the recombinant protein using streptag-based affinity chromatography in a StrepTrapTM HP 5 ml column (Cytiva), treated it with 5 mM Tris(2-carboxyethyl)phosphine hydrochloride (Thermo Fisher Scientific) for 10 min at room temperature to reduce all exposed cysteines and purified the protein using size-exclusion chromatography with

a Superdex 75 column (Cytiva) using SEC buffer (Tris–HCl 10 mM pH 8, NaCl 100 mM). The final yields obtained were: sF13$^{WT}$ = 3.5 mg l$^{-1}$, sF13$^{A295E}$ = 2.5 mg l$^{-1}$, sF13$^{G277C}$ = 2.8 mg l$^{-1}$, sF13$^{4MUT}$ = 1.4 mg l$^{-1}$, sF13$^{Δ267}$ = 0.6 mg l$^{-1}$, sF13$^{R291E}$ = 1 mg l$^{-1}$, sF13$^{S292F}$ = 1.9 mg l$^{-1}$, sF13$^{S292K}$ = 0.2 mg l$^{-1}$ and sF13$^{L296Y}$ = 0.4 mg l$^{-1}$. All proteins were analysed by SDS–PAGE to assess their purity (Extended Data Fig. 8a).

## Crystallization and structure determination

For crystallization, we digested the purification tags using 1.5 units of thrombin (Cytiva) per 0.1 mg of protein overnight at 4 °C and then treated the protein with Tris(2-carboxyethyl)phosphine hydrochloride for 10 min at room temperature. The digest was loaded to a gel filtration Superdex 75 16/60 column in 10 mM Tris–HCl pH 8.0, 100 mM NaCl, and the fractions of the main peak were pooled and concentrated to 12 mg ml$^{-1}$ in the same buffer for crystallization trials. Crystallization screening trials were carried out by the vapor diffusion method using a Mosquito TM nanodispensing system (STPLabtech) following established protocols[36]. Monoclinic crystals of sF13$^{WT}$ were grown after 10 days in 20% ($w/v$) PEG 3350, 0.1 M HEPES pH 7.5 and 2% ($v/v$) Tacsimate and were cryoprotected in the same solution supplemented with 20% ($v/v$) glycerol. Cubic crystals of sF13$^{WT}$ were grown in 1 day in 1 M Na$_3$ citrate and 0.1 M imidazole pH 8 and were cryoprotected using the crystallization solution supplemented with 33% ($v/v$) glycerol. To obtain the complex with tecovirimat, we soaked cubic crystals, which have a high solvent content (70%), for 5 min into a soaking solution containing 1 mM tecovirimat (BenchChem, catalogue number B611274), 10% ($v/v$) DMSO, 1 M Na$_3$ citrate and 0.1 M imidazole pH 8. To obtain the complex with IMCBH, we soaked cubic crystals into a solution containing 1 mM IMCBH (BLD Pharmatech, catalogue number BL3H9998EC8C), 10% ($v/v$) DMSO, 1 M Na$_3$ citrate and 0.1 M imidazole pH 8. After soaking, all crystals were cryoprotected using the soaking solution supplemented with 33% ($v/v$) glycerol. Similarly, cubic crystals of sF13$^{A295E}$ were obtained in 1 day using 1 M Na$_3$ citrate and 0.1 M imidazole pH 8 and soaked with tecovirimat as reported above.

X-ray diffraction data were collected on beamlines PROXIMA-1 and PROXIMA-2A at the synchrotron SOLEIL (St Aubin, France) using the beamline control software MXCuBE (version 2). Diffraction images were integrated with XDS (version 10 January 2022)[37], and crystallographic calculations were carried out with programs from the CCP4 program suite (version 9)[38]. To determine the phases, we used a model of F13 obtained using AlphaFold2 (ref. 39) as a template to perform molecular replacement in PHASER (version 2.8.3)[40]. To obtain the final models, we iteratively built and refined the structures using phenix.refine (Phenix version 1.19.2-4158)[41] and Coot (version 0.9.8.95)[42] using isotropic B factor and Translation/Libration/Screw groups as refinement strategy. We validated all the models using MolProbity (version 4.5.2)[43]. The crystallographic statistics are provided in Supplementary Table 1. In crystals soaked with tecovirimat or IMCBH, additional electron density appeared at the dimeric interface that was compatible with the shape and size of the drug, as shown in Fig. 2. To facilitate the modelling, all maps derived from the cubic crystals were corrected using a bulk-solvent mosaic model available in the PHENIX program (phenix.mosaic). The electron density for tecovirimat does not show clear features. We hypothesize that this is because the binding pocket is symmetric, while the molecule itself is not, allowing it to adopt two indistinguishable orientations rotated by 180°. However, it is also possible that this is a crystallographic artefact, with tecovirimat only entering in a single orientation, and the density is featureless because of the presence of a twofold symmetry axis crossing the molecule. To investigate this, we reprocessed the cubic crystals in the space group P1 and refined the tecovirimat molecule in two different ways: in a single orientation with 100% occupancy and in two orientations rotated by 180° with 50% occupancy each, mimicking what is observed in the cubic space group. When comparing the two refinements (Extended Data Fig. 8b), we observed improved R factors and reduced residual

densities when using the model with two rotated molecules, supporting our original hypothesis. Coordinates and structure factors have been deposited in the Protein Data Bank. Figures showing the crystallographic models were generated with PyMol v3.0.3 (Schrödinger, LLC).

## Molecular dynamics simulations

To assess the stability of the X-ray resolved F13 dimer on the membrane surface, we conducted molecular dynamics simulations. Before simulation, we made several structural modifications to the sF13 dimer. First, the unresolved N-terminal residues (residues 1–5) were modelled as unstructured and integrated into the dimer structure using the Modeller (version 10.4) tool[44]. Next, the structure was processed using the CHARMM-GUI (accessed on December 2023)[45] server to add post-translational palmitoylation on residues C185 and C186 and neutral capping of the N- and C-terminal residues. Subsequently, the F13 dimer was positioned on a membrane (Fig. 1c) mimicking the lipid composition of the Golgi membrane (as outlined in Supplementary Table 2). For the protein force field, we used the CHARMM36m-WYF force field[46,47], which includes corrections for cation–pi interactions, while lipids were described using the CHARMM36 force field[48]. The protein–membrane system was solvated with 52,801 water molecules using CHARMM-modified TIP3P[49] water model, and the total charge of the system was neutralized by adding 82 K$^+$ ions. The total system size is 249,131 atoms. The box dimensions were 14.34 × 14.34 × 13.56 nm in the $x$, $y$ and $z$ directions. The solvated system was energy minimized using the steepest descent algorithm to remove any steric clashes, followed by six short equilibrations ranging from 125 ps to 500 ps with restraints on either protein backbone/side chain atoms or lipid phosphate atoms.

Throughout the equilibration process, we maintained a temperature of 310 K using the Berendsen thermostat[50] with a time constant ($\tau_t$) of 1 ps, while pressure was maintained at 1 bar using the Berendsen semi-isotropic scheme with a time constant ($\tau_p$) of 5 ps. van der Waals and electrostatic interactions were treated using the cut-off and Particle Mesh Ewald[51,52] methods, respectively, with a cut-off of 1.2 nm. Covalent bonds involving hydrogen atoms were constrained using the LINCS algorithm[53]. For the final production run, we removed all restraints and switched to V-rescale[54] and Parrinello–Rahman[55,56] semi-isotropic scheme to regulate the temperature and pressure, respectively. The rest of the parameters were consistent with those used during equilibration. The production simulations were conducted for 1 μs with 5 repeats using the GROMACS (version 2021) simulation package[57], using a time step of 2 fs. For analysis, we concatenated the last 300 ns from each repeat and examined monomer–monomer contacts within 5 Å. All images and plots were generated using VMD (version 1.8.1)[58] and the matplotlib (version 3.10)[59] library.

Although the structure of the sF13 dimer was determined at a resolution of 2.6 Å, a crystallographic 2-fold axis passes through the ligand density (tecovirimat), leading to challenges in accurately fitting the ligand. To enhance the accuracy of ligand modelling within the density, we used the recently developed RosettaEMERALD (rosetta release-362)[60] protocol. This protocol integrates both RosettaGenFF and genetic algorithm optimization for robust ligand modelling within the density map. The three-dimensional structure of tecovirimat was downloaded from PubChem[61] in its endo-isomeric form, which is the favoured product of the Diels–Alder reaction used to synthesize the drug. Next, the sF13 dimer–tecovirimat complex was docked to the density using the ChimeraX (v1.7.1) tool[62]. Following this, we used the RosettaEMERALD protocol to accurately model tecovirimat within the density. Briefly, an initial pool of 500 ligand conformations, along with protein side chains, undergo genetic algorithm optimization over 10 generations. The top 20 lowest-energy conformations obtained from genetic algorithm optimization were further refined, along with protein side chains, using a cartesian minimization in Rosetta. The protocol was executed in triplicate. Out of the total 60 ligand conformations, redundant poses were eliminated, and 15 poses were selected for binding free

energy calculations. These 15 selected poses are indicated by the first number of *x*-axis labels of Extended Data Fig. 3. The RosettaEMERALD XML script and flags used for refining the ligand within the density are provided in the supplementary material.

## ABFE calculation

The selected 15 poses were subjected to binding free energy estimation using an in-house pipeline (publication in preparation). Our ABFE protocol is similar to the one previously described by ref. [63]. ABFE calculations were performed in triplicate for each pose. To optimize the ABFE calculations, the membrane was excluded from the simulation. This simplification is justified as the tecovirimat binding pocket is located at 40 Å from the membrane surface, and equilibration simulations show that the interface remains stable throughout the simulation (Extended Data Figs. 9 and 10). The AMBER-ff14sb[64] parameters for F13 dimer were acquired through the use of OpenMM (version 8.0)[65] and ParmEd (version 4.1.0)[66] software, while the OpenFF-2.0.0 (ref. [67]) parameters with AM1-BCC charges for tecovirimat were obtained via TOFF[68] software v0.1.0. The TIP3P[49] water model was used, along with AMBER parameters for ions. GROMACS-2022.4 (ref. [57]) simulation package was used as the molecular dynamic engine.

In all cases, the simulation temperature was maintained at 298.15 K using Langevin dynamics with a collision frequency of 2 ps$^{-1}$. Van der Waals and electrostatic interactions were treated using the cut-off and Particle Mesh Ewald methods[51,52], respectively, with a cut-off of 1 nm. Hydrogen bonds were constrained using the LINCS algorithm[53]. Two different isotropic schemes were used to maintain the pressure at 1 atm: Berendsen[50] with a time constant of 1 ps and Parrinello–Rahman[55,56] with a time constant of 2 ps. All production simulations used the former. We used a hydrogen mass repartitioning factor of 2.5, which allowed an integration time step of 4 fs for all production simulations. Other molecular dynamics parameters are detailed in the GROMACS input files provided as Supporting Information.

Our in-house pipeline operates as follows (Supplementary Fig. 1): During the 'Build Simulation System' phase, the user provides configuration details for the protein–ligand complex, and topologies are generated accordingly. Two neutral solvated systems with 150 mM of NaCl are created for the ligand and protein–ligand complex in an octahedron box with 1.5 nm distance between the solute and the edges' box with the GROMACS' solvate module. The 'Equilibration Setup' and 'Equilibration Run' steps generate molecular dynamics parameters and conduct the corresponding equilibration simulations. For the protein–ligand complex, the process begins with minimization using the steepest-descent algorithm. This is followed by a 1 ns NVT (constant particle number, volume and temperature) phase with a 2 fs integration time step and position restraints on the heavy atoms using a force constant of 2,500 kJ mol$^{-1}$ nm$^{-2}$. Next, a 1.05 ns NVT phase and approximately 1 ns NPT (constant particle number, pressure and temperature) phase are conducted, both with a 3 fs integration time step and the same position restraints. The previous NPT phase uses the Berendsen scheme as detailed above. Subsequently, a 5 ns NPT phase with the Parrinello–Rahman scheme with a 4 fs integration time step is performed without restraints. This is followed by a final step of 10 ns under the same conditions. For the ligand alone, the same procedure is followed, except the initial 1 ns NVT phase with a 2 fs integration time step is omitted and the final NPT simulation is performed during 5 ns. The final 10 ns of the of the protein–ligand complex simulation was used to estimate the optimal Boresch restraints for the decoupling phase of the protein–ligand complex simulations.

Similar to the previous two steps of the workflow, 'FEP Setup' and 'FEP Run' prepare and execute the FEP simulations needed to complete the thermodynamic cycle detailed in the 'Thermodynamic cycle for FEP' section. Each window for both the protein–ligand complex and the ligand alone begins with minimization using the steepest-descent

algorithm. This is followed by a 10 ps NVT phase with a 2 fs integration time step and position restraints on the heavy atoms using a force constant of 2,500 kJ mol$^{-1}$ nm$^{-2}$. Next, a 100 ps NPT phase is conducted with a 4 fs integration time step, the same position restraints and the Berendsen scheme as detailed above. Subsequently, a 500 ps NPT phase with the Parrinello–Rahman scheme and a 4 fs integration time step is performed without restraints. This is followed by a final step of 10 ns under the same conditions.

The free energy contributions of each step are computed using either the multistate Bennett acceptance ratio[69] or thermodynamic integration estimators during the 'Get Contribution' step. The Python package alchemlyb-2.0.0 (ref. [70]) was used for this purpose. Finally, all results are aggregated in the 'Get Cycle's $\Delta G$' step.

## Thermodynamic cycle for FEP

The thermodynamic cycle involved decoupling the Coulomb interactions of the ligand in water over 11 λ points, followed by the decoupling of van der Waals interactions over 21 λ points with a soft-core potential activated to prevent numerical instability. Boresch restraints[71], chosen from the last 10 ns of the protein–ligand complex free simulation during the equilibration phase ('ABFE calculation' section), had their free energy contribution analytically calculated.

Both the selection and energy contribution estimation of Boresch restraints were conducted using the software MDRestraintsGenerator (version 0.2.0)[72]. The selected restraints were activated for the ligand in complex with the protein, and the van der Waals interactions of the ligand were reactivated in the protein complex over 21 λ points with a soft-core potential to avoid numerical instability. Subsequently, Coulomb interactions were activated over 11 λ points to finally remove the restraints over 12 λ points. The binding free energy is calculated from the contributions of all previously mentioned steps.

## Clustering and identification of the most favourable energetic pose

Frames selected by MDRestraintsGenerator and used as input structures for the FEP simulations were clustered based on the protein–ligand interaction fingerprint calculated with ProLIF (version 2.0.3)[73]. Each bit of the fingerprint represents a pair of atom/atom groups from the protein and ligand involved in a specific class of interaction as defined by ProLIF. This unambiguous definition allows for the separation of potential poses that may be symmetrical. The final fingerprint consists of 956 bits. The similarity among frames is calculated using the Tanimoto metric implemented in RDKit (version 2023.03.2)[74] on the constructed protein–ligand interaction fingerprint, resulting in the generation of a similarity matrix.

The similarity matrix was then subjected to the hierarchical clustering algorithm in SciPy using Ward's variance minimization algorithm[75]. After constructing the dendrogram, the number of clusters was determined through visual inspection.

## ABFE between the most energetic favourable pose and the individual monomers

To investigate whether tecovirimat can bind to the monomer, the ABFE for each individual monomer was calculated for the identified most energetic favourable pose within the dimer. The same methodology previously described was used, with the only difference being that a single monomer was used instead of the dimer.

## Calculation of average binding free energies

To estimate the average binding free energy for the dimer and monomer complexes with tecovirimat, we averaged over all independent simulations ($N = 45$ for the tecovirimat–dimer, $N = 6$ for the tecovirimat–monomer binding) as shown in equation (1), implying that we average with respect to the binding probabilities (rather than with respect to the binding free energies). Thereby, the average is dominated

by the high-affinity binding poses. Here, $\beta$ is the inverse temperature, and $\langle \cdot \rangle$ denotes the average over independent simulations.

$$\Delta G_{\text{bind}} = -\beta^{-1}\ln\langle e^{-\beta\Delta G_{\text{bind},i}}\rangle \tag{1}$$

The 95% confidence interval was estimated from 1,000 rounds of bootstrapping. In each round, $N$ $\Delta G_{\text{bind}_i}$ samples were drawn with replacement from our $N$ $\Delta G_{\text{bind}_i}$ values and averaged according to equation (1). After removing the largest and smallest 2.5% of the 1,000 bootstrapped averages, 95% confidence intervals were obtained from the upper and lower bounds of the remaining 950 averages.

## Validation of ABFE calculation via ABFE calculations for seven additional ligands

To provide additional evidence for the tecovirimat binding pose and to validate the ABFE calculations, we carried out additional ABFE calculations with seven structurally similar ligands with available $EC_{50}^{\text{VACV}}$ values alongside tecovirimat[22,23]. Here $EC_{50}^{\text{VACV}}$ denotes the effective concentration that inhibits 50% of virus-induced cytopathic effects on VACVs. Each ligand was aligned to the tecovirimat pose reported here, and ABFE calculations were conducted using three independent replicates for each of the eight ligands, including tecovirimat. The calculated binding affinity $\Delta G_{\text{calc}}$ reported for each ligand represents the mean across the three replicates, and the error was taken as the standard error of the mean (s.e.m.).

We assume that the $EC_{50}^{\text{VACV}}$ value is related to the change in free energy upon two reactions, dimerization and ligand binding:

$$2P \rightleftharpoons PP^{\text{X-ray}}; \Delta G^{\circ}_{\text{dimer}}$$

$$PP^{\text{X-ray}} + L \rightleftharpoons PPL^{\text{X-ray}}; \Delta G^{\circ}_{\text{complex}} \tag{2}$$

$$2P + L \rightleftharpoons PPL^{\text{X-ray}}; \Delta G^{\circ}_{\text{dimer}} + \Delta G^{\circ}_{\text{complex}}$$

where $P$ denotes the protomer (a single monomer), $L$ the ligand, $PP^{\text{X-ray}}$ the homodimer observed in the crystal, and $PPL^{\text{X-ray}}$ is the ternary complex. $\Delta G^{\circ}_{\text{dimer}}$ denotes the free energy for dimerization of two protomers towards the dimeric crystal structure, and $\Delta G^{\circ}_{\text{complex}}$ denotes the free energy for ligand binding to the crystallographic homodimer. For the overall reaction, the fraction of protein in ternary complex is

$$\theta = \frac{2a_{\text{PPL}}}{a_{\text{P}} + 2a_{\text{PPL}}} \tag{3}$$

Here, $a_i$ denotes the activity of species $i$ defined as $a_i \equiv C_i/C^{\circ}$, where $C^{\circ}$ is the standard concentration of 1 mol l$^{-1}$. The dissociation constant for equation (2) is:

$$K_d = \frac{a_{\text{P}}^2 a_{\text{L}}}{a_{\text{PPL}}} \tag{4}$$

Equations (3) and (4) yield

$$K_d = \frac{2(1-\theta)}{\theta} a_{\text{P}} a_{\text{L}} \tag{5}$$

Let $a_L^*$ denote the ligand activity at which 50% of the protein is in complex ($\theta = 0.5$). Here '*' is used to distinguish the symbol $a_L$, the activity of the ligand at any $\theta$ value. Thus, we have:

$$K_d = 2a_{\text{P}}a_L^* \tag{6}$$

Assuming that $a_L^*$ is proportional to $EC_{50}^{\text{VACV}}$ among the eight ligands, we have $a_L^* = \gamma EC_{50}^{\text{VACV}}/C^{\circ}$, where $\gamma$ is an unknown constant. Thus,

$$\Delta G^{\circ}_{\text{complex}} + \Delta G_{\text{dimer}} = RT\ln(K_d) = RT\ln\left(\frac{EC_{50}^{\text{VACV}}}{C^{\circ}}\right) + RT\ln(2\gamma a_{\text{P}}), \tag{7}$$

where $R$ is the gas constant and $T$ the temperature. Furthermore, we assume that the activity of the protomer $a_{\text{P}}$ on the cell is constant.

In our ABFE calculations, we evaluated only the second step of the two reactions of equation (2):

$$PP^{\text{X-ray}} + L \rightleftharpoons PPL^{\text{X-ray}}; \Delta G^{\circ}_{\text{calc}} \tag{8}$$

Thus, $\Delta G^{\circ}_{\text{calc}}$ is offset from $RT\ln\left(\frac{EC_{50}^{\text{VACV}}}{C^{\circ}}\right)$ by two constant contributions:

$$\Delta G_{\text{offset}} = -\Delta G^{\circ}_{\text{dimer}} + RT\ln(2\gamma a_{\text{P}}) \tag{9}$$

$\Delta G_{\text{offset}}$ accounts for (1) the free energy cost of F13 dimerization and (2) our assumption that the intracellular ligand activity is proportional (but not equal) to extracellular ligand concentration in experiment. By comparison of our calculated AFBEs with the experimental $EC_{50}^{\text{VACV}}$ values, we estimated $\Delta G_{\text{offset}} \approx \langle\Delta G^{\circ}_{\text{calc}} - \Delta G^{\circ}_{\text{exp}}\rangle = -15.2 \pm 0.4$ kcal mol$^{-1}$, where $\langle \cdot \rangle$ denotes the average over the eight ligands. The error for the quantity $RT\ln\left(\frac{EC_{50}^{\text{VACV}}}{C^{\circ}}\right) + \Delta G_{\text{offset}}$ was estimated by error propagation using the uncertainties Python library[76].

Supplementary Fig. 1 correlates $\Delta G^{\circ}_{\text{calc}}$ with $RT\ln\left(\frac{EC_{50}^{\text{VACV}}}{C^{\circ}}\right)$, after correcting for $\Delta G_{\text{offset}}$. The reasonable agreement (1) validates the ABFE protocol and (2) suggests that the crystallographic pose of tecovirimat is adopted by the other seven ligands considered in this analysis.

## Mass photometry
Mass photometry experiments were done using TwoMP instrument (Refeyn) using filtered (0.22 μm) 'protein buffer' (10 mM Tris–HCl pH 8, 100 mM NaCl) to avoid contaminations which would increase the background signal. Contrast-to-mass calibrations were achieved by measuring the contrast of two references (bovine serum albumin (BSA) and urease, both purchased from Sigma Aldrich) diluted in protein buffer, covering mass range from 66 kDa to 272 kDa. Four contrast values were used to generate a standard calibration curve, with the following rounded average masses: 66, 132, 198 and 272 kDa. We performed the experiment using microscope coverslips (24 × 50 mm and 170 ± 5 μm thick) cleaned with isopropanol and Milli-Q water followed by drying with air. Samples were loaded into dried coverslip surface assembled into silicone gaskets. Immediately before mass photometry measurements, 2 μl of sF13 protein stocks, with increasing amounts of tecovirimat or IMCBH, was diluted in 18 μl of 'protein buffer' into the gasket hole and mixed twice. In all cases, the final concentration of sF13 was 25 nM. Tecovirimat/IMCBH were at different concentrations between 10 μM and 1 nM. Data acquisition was performed using AcquireMP v2.3 (Refeyn), and movies of 2,936 frames were recorded at 49 Hz framerate, adjusted to maximize camera counts while avoiding saturation. Mass photometry images were processed and analysed using DiscoverMP v2.3 (Refeyn).

Mean contrast values from the BSA and urease calibration were plotted and fitted to a linear function $y = bx$, where y is the contrast, $x$ is the mass and $b$ is the contrast-to-mass calibration factor. To extract mole fractions (percentage of each species), we converted all particle contrasts obtained from each movie to mass, applied a Gaussian fitting and calculated mole fractions as the area of each Gaussian curve. Finally, sF13 dimer percentage values were plotted against tecovirimat/IMCBH concentration using Prism Graphpad v9.0.2, and EC50 values were extracted using a nonlinear fit function (Extended Data Fig. 6).

## AUC
Sedimentation velocity experiments were carried out at 20 °C in an Optima AUC analytical ultracentrifuge (Beckman Coulter) equipped with double-UV and Rayleigh interference detection. Purified sF13 proteins at 0.4 mg ml$^{-1}$ in the presence or absence of tecovirimat (10 μM) were centrifuged at 42,000 r.p.m. (23,600 g) using an AN60-Ti rotor and 12 mm thick double sector centrepieces. Absorbance and interference

profiles were recorded every 5 min. Buffer viscosity ($\eta = 1.016$ cP) and density ($\rho = 1.0054$ g ml$^{-1}$) at 20 °C were estimated with SEDNTERP 1.09. Partial specific volumes at 20 °C were estimated based on amino acid sequences using SEDNTERP 1.09 software. Data were analysed with SEDFIT 16.1 (ref. 77) using a continuous size distribution c(S) model. Theoretical sedimentations of the complex were generated using hydropro 10 (ref. 78).

## SAXS experiments

SAXS data were collected on the SWING beamline at Synchrotron Soleil (France) using the online HPLC system. These experiments have been performed using sF13$^{WT}$ digested with thrombin. sF13$^{WT}$ samples at 4.6 mg ml$^{-1}$ were prepared in a buffer containing 10 mM Tris pH 8, 100 mM NaCl and 10 μM tecovirimat and injected into a size exclusion column (Superdex 75 increase 5/150 mm) cooled at 15 °C eluting directly into the SAXS flow-through capillary cell at a flow rate of 200 μl min$^{-1}$. The data were analysed using FOXTROT and PRIMUS from ATSAS 3.2 (ref. 79), from which Guinier was generated. Scattering curves were selected for stable $R_g$ at the apex of the elution profile, the selected curves were averaged, and buffer signal was subtracted. From these corrected scattering curves, the pair distribution function was computed using GNOM (version 5.0)[80], and the normalized Kratky plot was generated. Using the structure of sF13$^{WT}$ (PDB 9FHS), the experimental curve was compared to theoretical curve using CRYSOL (version 2.8.3)[81]. Ab initio models were generated with DAMMIN (version 5.3)[82], and for each model, sedimentation characteristic was calculated with hydropro (version 10)[78]. The SAXS statistics are provided in Supplementary Tables 6 and 7.

## F13 transfection for PLA and immunofluorescence staining

To perform the PLA experiment and immunofluorescence staining of F13, HeLa cells (ATTC CCL-2) were transfected with pcDNA 3.1 plasmids coding for either F13$^{WT}$ or F13$^{4MUT}$ (N267D, A288P, A290V, D294V), with an internal FLAG tag sequence (GGGDYKDDDDKGGG) inserted within residues D21 and N22. The use of an internal FLAG tag in F13 was necessary, as the N-terminus is buried into the membrane and the C-terminus is part of the dimeric interface. Thus, none of them were suitable for standard N- or C-terminal protein tagging. We selected the best region to insert the FLAG tag based on the sF13 dimer structure reported here. For this, we selected an exposed loop, away from the membrane interaction region and the dimerization interface.

For PLA and immunofluorescence, $1.2 \times 10^4$ HeLa cells per well were transfected in suspension using lipofectamine 2000 (Thermo Fisher Scientific) in a 96-well plate (μClear, Greiner Bio-One 655090) with 100 ng of DNA. In each well 50 μl of HeLa cells at $2.4 \times 10^5$ cells ml$^{-1}$ were mixed with 50 μl of transfection mix and 50 μl of DMSO or DMSO/tecovirimat, resulting in tecovirimat at a final concentration of 10 μM and DMSO at 0.1%. Cells were incubated 24 h at 37 °C and 5% CO$_2$; subsequently the cells were fixed with 4% PFA for 10 min.

## PLA

PLA was performed using the Duolink PLA Fluoresence kit (DUO92008, Merck). In short, cells were permeabilized at room temperature for 3 min in PBS with Triton 0.1% and washed with PBS. About 40 μl of Duolink blocking solution was added to each well, and the plate was incubated at 37 °C for 1 h. After blocking, cells were incubated at room temperature for 45 min with primary monoclonal mouse M2 (dilution = 1:350, F3165, Sigma-Aldrich) and rabbit D6W5B (dilution = 1:500, 14793, Cell Signaling Technology) anti-FLAG antibodies (diluted in Duolink Antibody Diluent) at a final concentration of 285 ng ml$^{-1}$. The wells were washed twice for 5 min at room temperature with buffer A (10 mM Tris pH = 7.4, 150 nM NaCl, 0.05% Tween). About 40 μl of PLA probe mix, containing PLA probe PLUS (anti-rabbit, dilution = 1:5, DUO92002, Merck) and PLA probe MINUS (anti-mouse, dilution = 1:5, DUO92004, Merck) was added to the wells following the manufacturer's

instructions. The plate was then incubated at 37 °C for 1 h. After incubation, the wells were washed twice for 5 min with buffer A, then 40 μl of ligase mix was added to the samples following the manufacturer's instructions; these were incubated 30 min at 37 °C. Wells were washed twice for 5 min with buffer A at room temperature; subsequently, 40 μl of amplification mix (containing a polymerase) were added to each well, following the manufacturer's instructions. The plate was then incubated for 100 min at 37 °C. Finally, PLA wells were washed once with buffer B (200 mM Tris, pH = 7.5, 100 mM NaCl) for 10 min at room temperature and a second time for 10 min at room temperature with buffer B supplemented with 1 μg ml$^{-1}$ Hoechst 33342 nuclear staining (Invitrogen). Subsequently, a final wash was performed with 0.01× buffer B for 1 min at room temperature, and cells were then left in fresh PBS. The plates were imaged using an Opera Phenix Plus microscope (Revvity) at ×20. Forty-nine images per well, covering over 90% of the well, were acquired.

## Immunofluorescence staining of F13

Immunofluorescence staining of F13 with rabbit and mouse anti-FLAG antibodies was performed in parallel with PLA, in the same plate. Cells were permeabilized at room temperature for 3 min in PBS with Triton 0.1% and washed with PBS. About 40 μl of Duolink blocking solution was added to each well, and the plate was incubated at 37 °C for 1 h. Cells were incubated at room temperature for 45 min with primary monoclonal mouse M2 (dilution = 1:350, F3165, Sigma-Aldrich) and rabbit D6W5B (dilution = 1:500, 14793, Cell Signaling Technology) anti-FLAG antibodies (diluted in Duolink Antibody Diluent) at a final concentration of 285 ng ml$^{-1}$. The wells were washed twice for 5 min at room temperature with buffer A. Wells were washed twice with PBS for 5 min at room temperature, and Alexa FluorTM 488 goat anti-mouse antibody (dilution = 1:500, A-11001, Invitrogen) and Alexa FluorTM 488 goat anti-rabbit antibody (dilution = 1:500, A-11008, Invitrogen) (diluted in PBS, BSA 1%, Na Azide 0.1%) were added to the respective wells at a final concentration of 4 μg ml$^{-1}$. The plate was then incubated at 37 °C for 1 h. Wells were washed once for 5 min at room temperature with PBS and once with PBS supplemented by 1 μg ml$^{-1}$ Hoechst 33342 nuclear staining (Invitrogen). Finally, cells were left in fresh PBS before imaging. The plates were imaged using an Opera Phenix Plus microscope (Revvity) at ×10. Twenty-one images per well, covering over 90% of the well, were acquired.

## Viral plaque assay and analysis

Six-well plates were seeded with BSC40 (ATTC CRL-2761) cells 24 h before infection. Confluent BSC40 cells were infected with wild-type VACV-WR[25] (provided by J.M. (University of Birmingham)) or rVACV mutants generated from a modified VACV-WR (vNotI/tk) strain (originally provided by B. Moss to K.C., who further modified it by incorporating an mCherry reporter into the tk locus) at a 10-fold dilution in DMEM with 2.5% FBS for 1 h at 37 °C. The infection medium was then removed, and a 0.5% methylcellulose in DMEM media overlay containing 10 μM Tecovirimat was added for 3 days at 37 °C. Afterward, the overlay medium was removed, and the wells were fixed and stained with 1% crystal violet in 20% methanol for 20 min. The crystal violet was removed, wells were washed with PBS, and plates were imaged using a Cytation 7. Plaque counts and diameters were measured to determine titres (plaque-forming units (p.f.u. ml$^{-1}$)) and plaque sizes (μm) using a program developed on the Cytation 7. The experiments with MPXV were done using the clade IIb strain MPXV/2022/FR/CMIP, which was isolated at the Institut Pasteur (France) in 2022. All experiments were conducted under struct BSL3 conditions according to the French regulations on dual use pathogens. The neutralization assays were done using Vero cells (ATTC CCL-81) plated in a μClear 96-well plate (Greiner Bio-One). The following day, cells were incubated with serial dilutions of Tecovirimat/IMCBH. MPXV was added 4 h later. The cells were fixed after 48 h with 4% paraformaldehyde and washed and immunostained with

polyclonal anti-VACV antibodies (PA1-7258, Invitrogen) and an Alexa Fluor 488-coupled goat anti-rabbit antibody (CA-11008, Invitrogen). Images were acquired with an Opera Phenix high-content confocal microscope (PerkinElmer). Infection was quantified by calculating the total area of MPXV-positive cells (MPXV+ area), and the nuclei were counted using the Harmony software v4.9 (PerkinElmer).

## Image analysis

PLA and immunofluorescence images were analysed using Signals Image Artist v1.3 (Revvity). For PLA (Extended Data Fig. 6), the PLA area (633 nm) was calculated using an intensity threshold. In parallel the number of cells was quantified using nuclear count on the Hoechst channel. The average PLA area per cell was calculated by dividing the total PLA area by the total number of nuclei per well. For quantification of the number of FLAG (F13) positive cells in each condition, first the number of cells was quantified using nuclear count on the Hoechst channel. Second, the number of nuclei positive for FLAG staining was calculated using an intensity threshold method on the nuclei region of interest. The percentage of FLAG+ cells was calculated by dividing the total number of FLAG+ nuclei by the total number of nuclei per well.

## Sequence analysis

We retrieved all MPXV genomes available on the GISAID database[26] and all OPXV available on the GenBank database as of 27 May 2024. For each viral species, sequences corresponding to the F13 coding sequence were extracted and aligned using MAFFT v7.505. Alignments were manually curated for accuracy using Geneious Prime v2024.0.5, and sequences covering less than 70% of the coding sequence were discarded. After these steps, we obtained a dataset comprising F13 sequences from 108 MPXV clade I, 8,472 MPXV clade II, 81 VARV, 211 VACV, 13 ECTV, 98 CPXV, 11 CMLP, 1 BRPV, 6 Akhmeta_virus (AKMV), 2 Orthopoxvirus Abatino, 2 Skunkpox virus (SKPV), 2 Taterapox virus (TATV) and 4 Raccoonpox virus (RCNV). Sequences were translated, and all variations to the consensus of each species were extracted. The mutations shown in Supplementary Table 3 were extracted from refs. 9,13,82–84. They list all mutations identified so far conferring resistance to tecovirimat. The potential escape mutants identified in this manuscript are described in ref. 27 and are available.

## Statistics and reproducibility

Data were collected from at least two independently repeated experiments, as indicated in the figure legends. Data collection and analysis were not performed blind to the conditions of the experiments. Data distribution was assumed to be normal, but this was not formally tested. No statistical method was used to predetermine sample size. No data were excluded from the analyses. Values are shown as mean ± s.d. Prism (GraphPad Software) was used to determine statistical significance. Two-way analysis of variance was used for analysis. *n* represents the number of independent samples.

## Reporting summary

Further information on research design is available in the Nature Portfolio Reporting Summary linked to this article.

## Data availability

Atomic coordinates of the reported structures have been deposited in the Protein Data Bank under accession codes 9FHK, 9FHS, 9HAH, 9FJ1, 9FIZ, 9FJA and 9FJ0. All molecular dynamics parameters, input topologies, coordinates, simulation control files, analysis scripts and files containing the sampled $\Delta H$ and $\Delta H/\Delta\lambda$ for ABFE are available via Zenodo at https://doi.org/10.5281/zenodo.14096216 (ref. 85). All the F13 sequences were extracted from the GISAID database at www.gisaid.org, and a summary of the accession numbers can be found in Supplementary Table 8. Source data are provided with this paper.

## Code availability

All molecular dynamics analysis scripts and files containing the sampled $\Delta H$ and $\Delta H/\Delta\lambda$ for ABFE are available via Zenodo at https://doi.org/10.5281/zenodo.14096216 (ref. 85).

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

## Acknowledgements

We thank the staff of the Institut Pasteur Crystallography facility for help with crystallization trials and the staff of the PX1, PX2 and SWING beamlines at synchrotron SOLEIL (St Aubin, France) for beamline support. We gratefully acknowledge the authors from the originating and the submitting laboratories, who generated and shared via GISAID genetic sequence data on which this research is based (Supplementary Table 8). We acknowledge support from the Institut Pasteur, the French National Research Agency (ANR; ANR-22-CE11-0003) to P.G.-C., the European Union's Horizon 2020 research, the innovation program under Marie Sklodowska Curie Grant 860592 and Deutsche Forschungsgemeinschaft (DFG, German Research Foundation; grants SFB1027/B7 and INST256/539-1) to J.S.H. The laboratory of E.S.-L. is funded by Institut Pasteur, the INCEPTION program (Investissements d'Avenir grant ANR-16-CONV-0005), the Ixcore foundation for research, the French Government's Labex Integrative Biology for Emerging Infectious Diseases (IBEID) (ANR-10-LABX-62-IBEID), the Health Emergency Preparedness and Response (HERA) Project DURABLE (grant number 101102733) and the National Institutes of Health (NIH) project Pasteur International Center for Research on Emerging Infectious Diseases (PICREID, grant number U01AI151758). The laboratory of O.S. is funded by Institut Pasteur, Urgence COVID-19 Fundraising Campaign of Institut Pasteur, Fondation pour la Recherche Médicale, Agence Nationale de Recherche sur le Sida et les Hépatites virales (ANR), the Vaccine Research Institute (ANR-10-LABX-77), Labex IBEID (ANR-10-LABX-62-IBEID), the HERA projects DURABLE (grant 101102733) and Integrating Multi-Disciplinary Expertise in a Learning and Adaptive European Pandemic Preparedness System (LEAPS). The funders had no role in study design, data collection and analysis, decision to publish or preparation of the manuscript.

## Author contributions

Conceptualization: K.C., O.S., J.S.H. and P.G.-C.; methodology and formal analysis: R.V., A.M.L., C.S.P., J.B., C.S., Y.J., A.M., F.G.-B., F.P., A.H., M.C., B.R., E.S.-L. and P.G.-C.; investigation: R.V., A.M.L., C.S.P., J.B., C.S., Y.J., J.S.H. and P.G.-C.; resources: J.M., K.C., O.S., P.G.-C.; writing: A.M.L., C.S., A.H., J.M., E.S.-L., K.C., O.S., J.S.H. and P.G.-C.; reviewing and editing: all authors; visualization: R.V., A.M.L., C.S.P., J.B., C.S. and P.G.-C.; supervision: P.G.-C., J.S.H., O.S. and K.C.; funding acquisition: P.G.-C., J.S.H., O.S. and K.C.

## Competing interests

The authors declare no competing interests.

## Additional information

**Extended data** is available for this paper at https://doi.org/10.1038/s41564-025-01936-6.

**Correspondence and requests for materials** should be addressed to Pablo Guardado-Calvo.

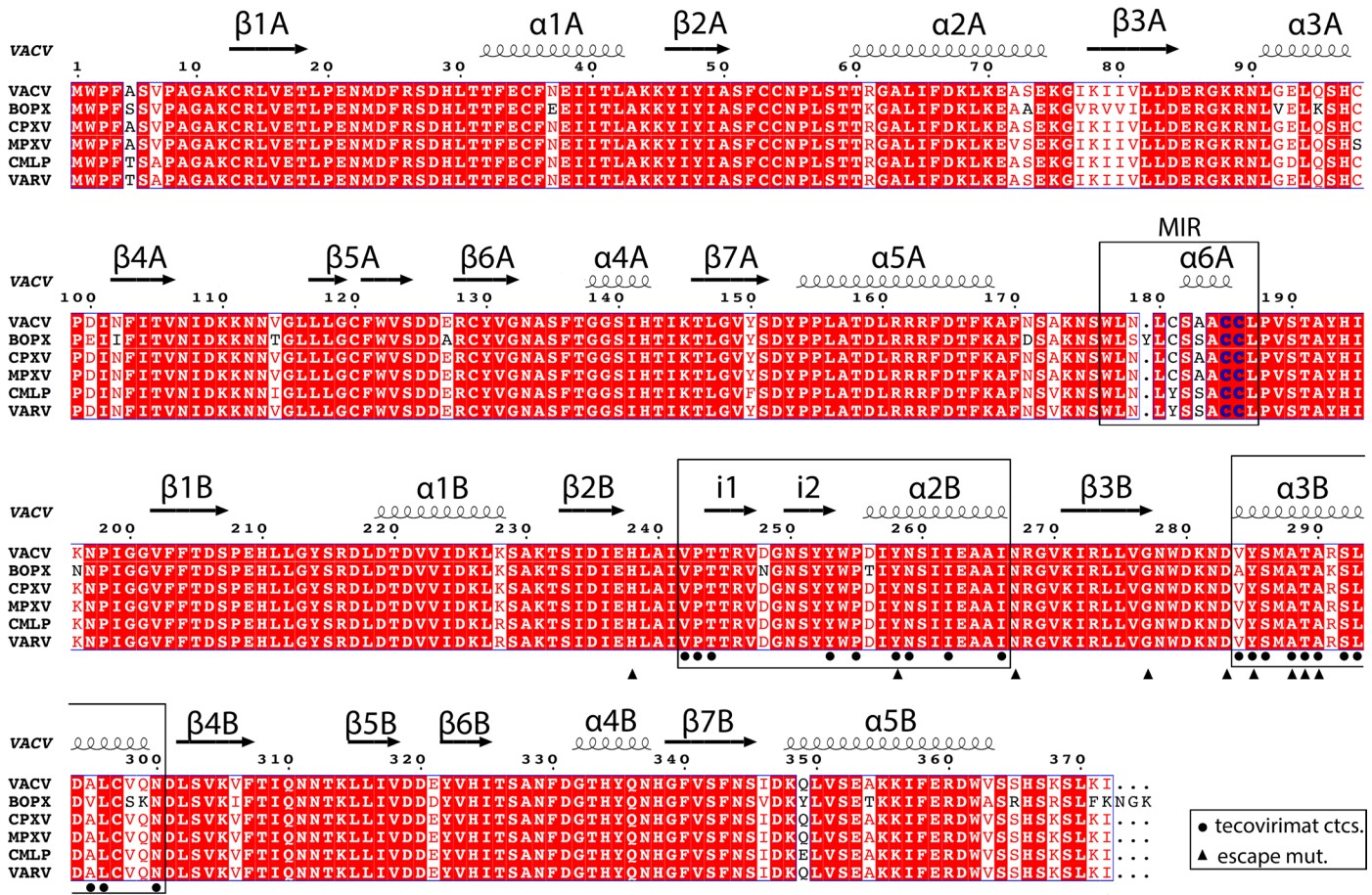

**Extended Data Fig. 1 | Multiple sequence alignment.** Multiple sequence alignment from six representative OPXVs. The secondary elements are indicated at the top. Strictly conserved residues are highlighted in red. The palmitoylated cysteines are shown in blue and the membrane interacting region (MIR) framed and labeled. Tecovirimat contacts and positions of escape mutants are marked with black circles and triangles under the alignment, as indicated. The accession codes for the F13 proteins used in the alignment are: Borealpox virus (BOPX, QED21148.1), Camelpox virus (CMLP, A0A0K1LD56), Variola virus (VARV, AAA60785.1), Vaccinia virus (VACV, P04021), Monkeypox virus (MPXV, YP_010377040.1), Cowpox virus (CPXV, CAD90601.1). the alignment was performed using clustal omega [89] and the figure prepared with ESPRIPT [90].

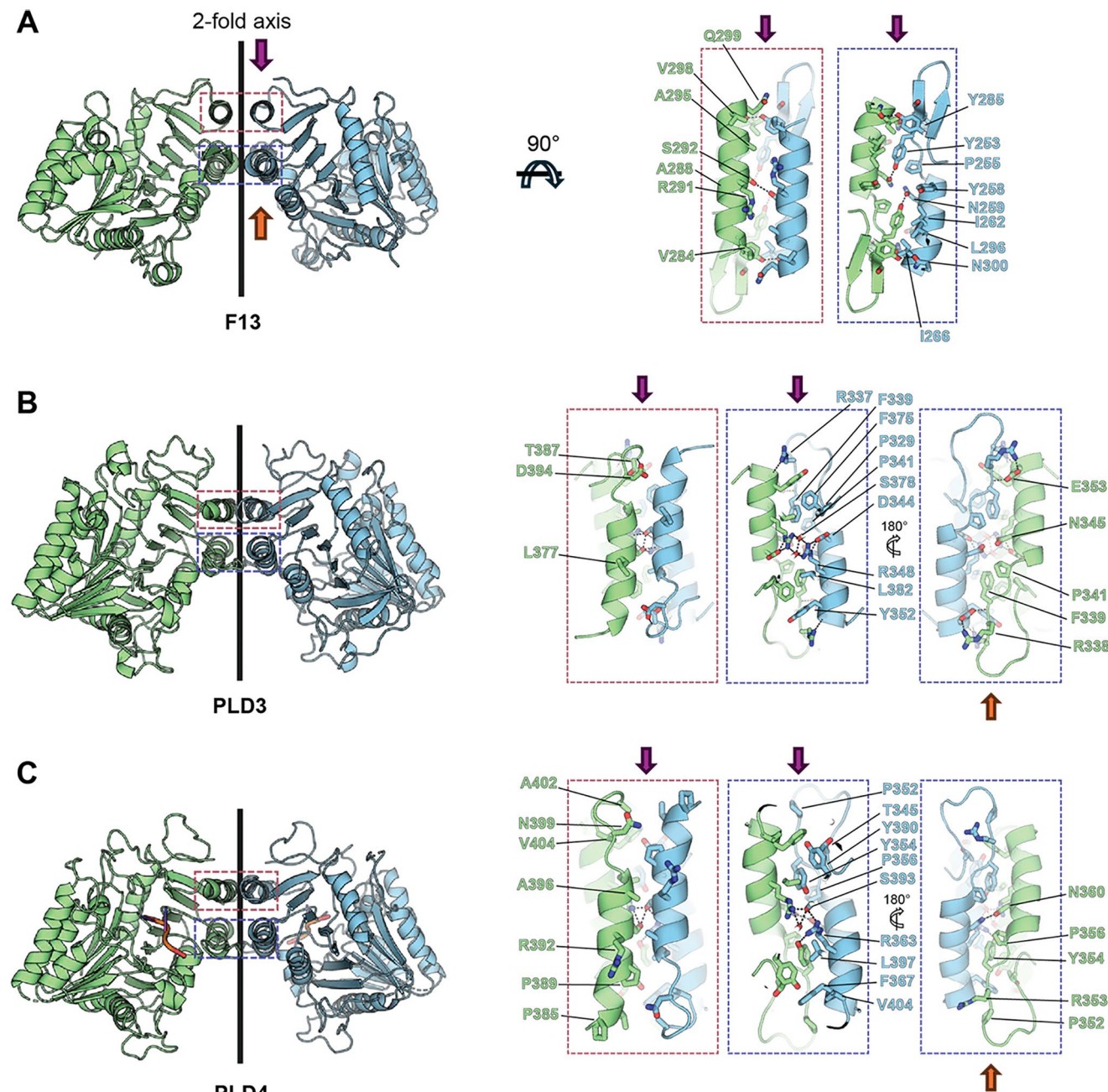

**Extended Data Fig. 2 | Contacts at the dimer interface. a**) The left panel shows the crystal structure of the sF13 homodimer (PDB: 9FHS) represented in cartoon form. One protomer is colored blue, and the other is green. The 2-fold axis is indicated by a black line. The right panels provide two close-up views of the dimer interface, as indicated in the left panel. The purple and yellow arrows indicate a top view or a bottom view of the dimer interface, respectively, as shown in the left panel. The main residues contributing to the dimer interface (identified using the PDBePISA server) are depicted as sticks and labeled (green or blue). Polar contacts are shown as dashed black lines. **b**) Similar to A, the left panel shows the crystal structure of the PLD3 homodimer (PDB: 8V05), with protomers colored green and blue. The right panels are close-up views of the dimer interface. **c**) Similar to A, the left panel shows the crystal structure of the PLD4 homodimer (PDB: 8V08), and the right panels are close-up views of the dimer interface.

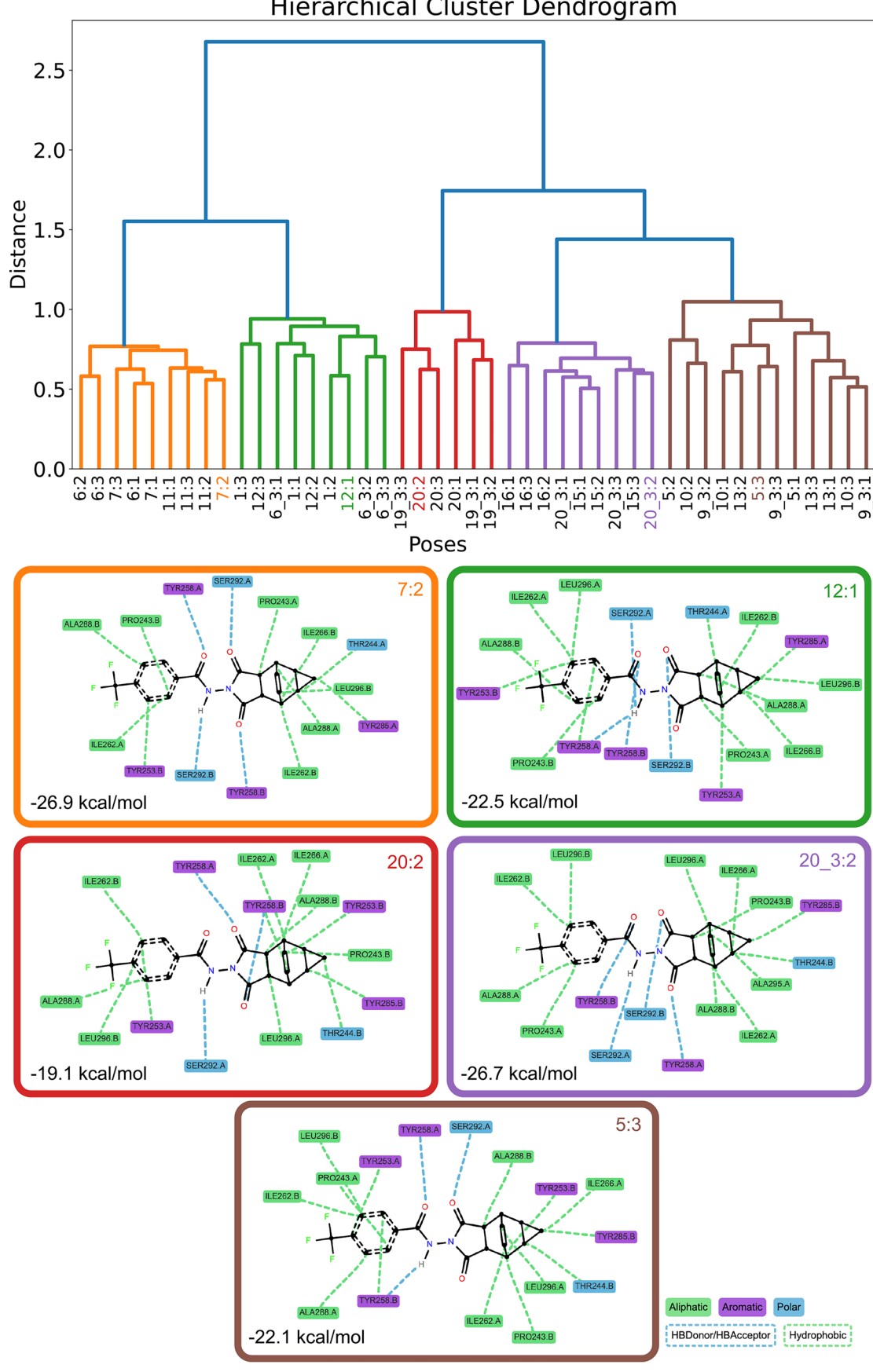

**Extended Data Fig. 3 | See next page for caption.**

**Extended Data Fig. 3 | Cluster analysis of the 45 input structures for free energy perturbation simulations.** Analysis of putative binding poses after equilibrating the 15 selected poses (x-label, first number) by three independent MD replicas (x-label, second number, see Methods). To reveal reoccurring protein-ligand interaction networks, a cluster analysis was performed. The x-axis represents the pose identification, and the y-axis indicates the distance between each pose in terms of their protein-ligand interaction network. The lowest energy pose of each cluster is highlighted, with its protein–ligand interaction profile and corresponding absolute binding free energy. The analysis demonstrates (together with Supplementary Table 4) that different protein-ligand interaction networks yield similar absolute binding free energies. The pose with the strongest binding affinity of -25.6 kcal/mol was used for further refinement against the crystallographic data, as reported in the PDB file. The Python library ProLIF[73] was used to generate the protein-ligand interaction network.

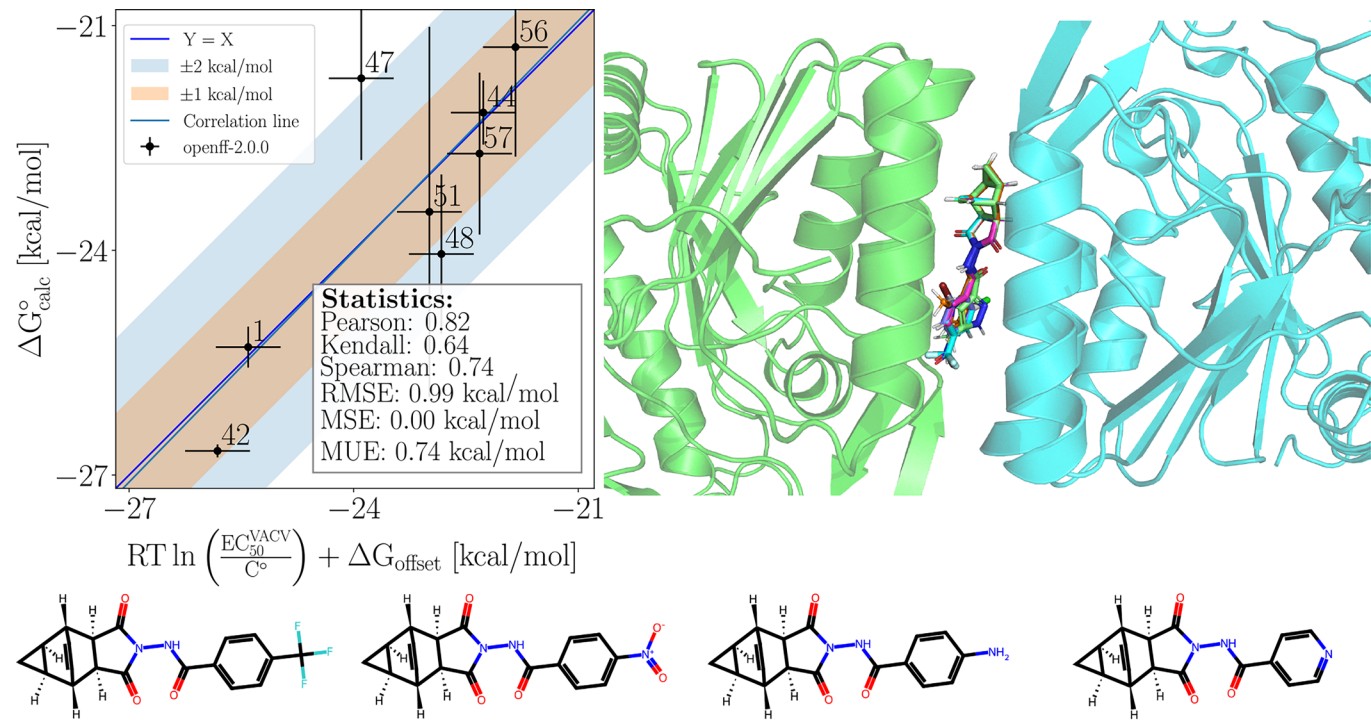

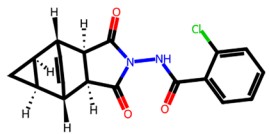

**1**: EC50(VACV) = 0.04 μM (Tecovirimat)

**42**: EC50(VACV) = 0.02 μM

**44**: EC50(VACV) = 7.7 μM

**47**: EC50(VACV) = 0.5 μM

**48**: EC50(VACV) = 3 μM

**51**: EC50(VACV) = 2.3 μM

**56**: EC50(VACV) = 15.8 μM

**57**: EC50(VACV) = 7.1 μM

**Extended Data Fig. 4 | Validation of the tecovirimat conformation using seven structurally similar ligands.** Eight ligands, including tecovirimat, were aligned to the proposed tecovirimat binding pose, and the absolute binding free energy for each ligand binding to the dimer was calculated (see Methods). Pearson, Kendall, and Spearman correlation coefficients are reported alongside root mean square error (RMSE), mean signed error (MSE), and mean unsigned error (MUE). Ligands are identified in the validation plot by numbers taken from[23]. Top right: 3D alignment of the eight molecules within the binding pocket. Bottom: chemical structures of the ligands with the corresponding EC50 values. The alignment was rendered with PyMOL[91] and chemical structures were drawn with RDKit[92].

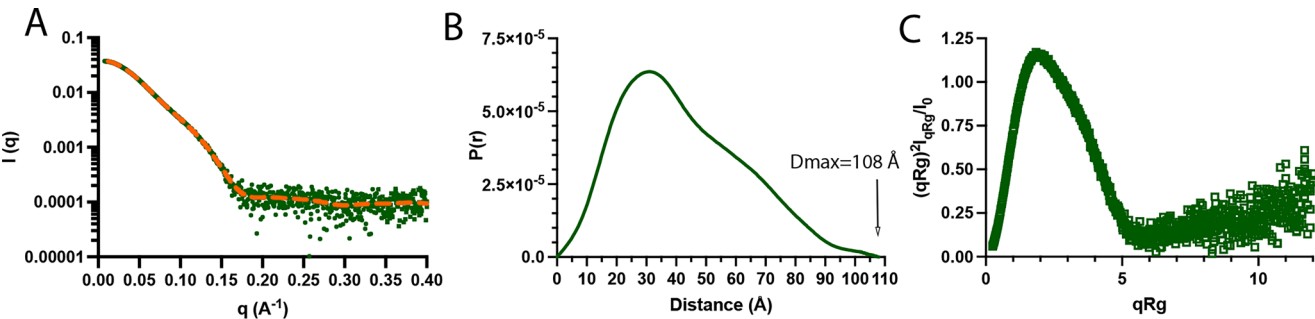

**Extended Data Fig. 5 | SAXS analysis of sF13/tecovirimat. a**) Guinier plot showing the experimental scattering curve of the F13/tecovirimat complex in green, and the fitted curve used to generate the pair distance distribution in orange. **b**) Pair distance distribution function calculated using GNOM[80] used to obtain Dmax and Rg values. **c**) Dimensionless (normalized) Kratky plot showing the characteristic shape of a well-folded protein.

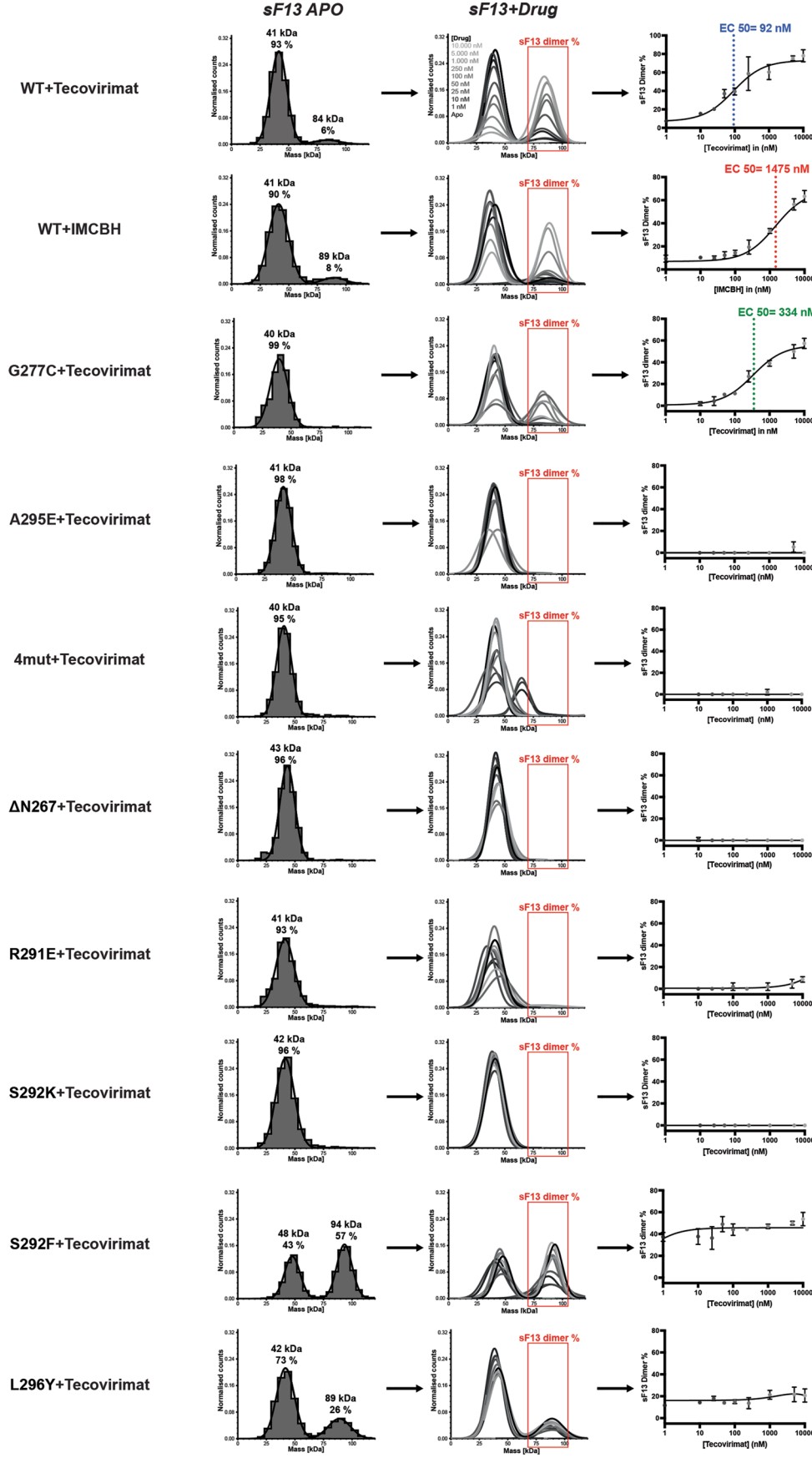

**Extended Data Fig. 6 | See next page for caption.**

**Extended Data Fig. 6 | Mass photometry (MP) assay.** Each row shows the pipeline used to determine EC50 values for sF13 wild-type and the different mutants, as indicated. The left column displays the mass distribution for sF13 at 25 nM without tecovirimat. The middle column shows mass distribution curves for sF13 with increasing drug concentrations, as indicated, highlighting in red the region used to calculate the percentage of dimers for the dose-response curve in the right column. The EC50 is derived from this dose-response curve. For clarity, the middle column presents a single representative experiment per drug concentration; however, the dose-response curve is based on the mean and standard deviation from three repeated experiments.

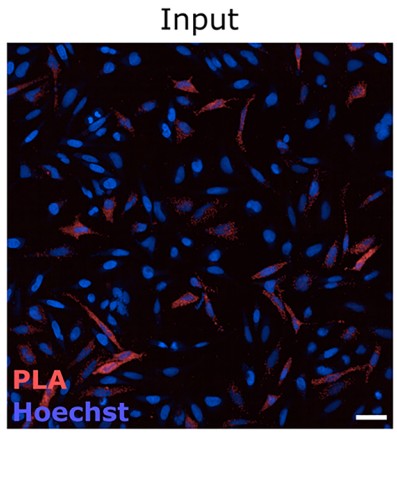
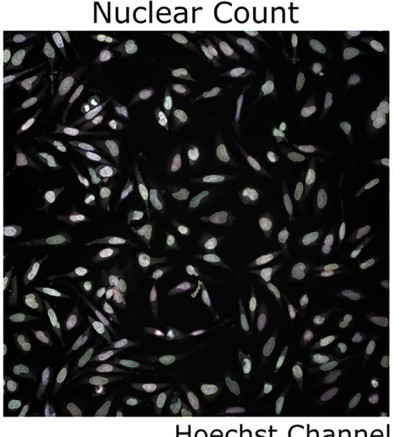
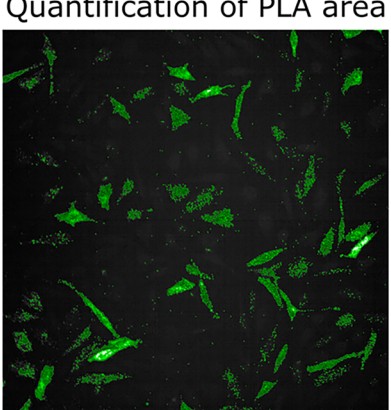

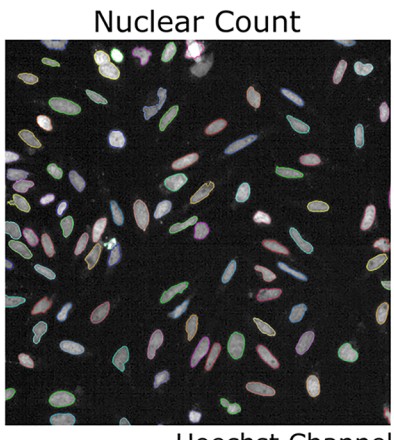

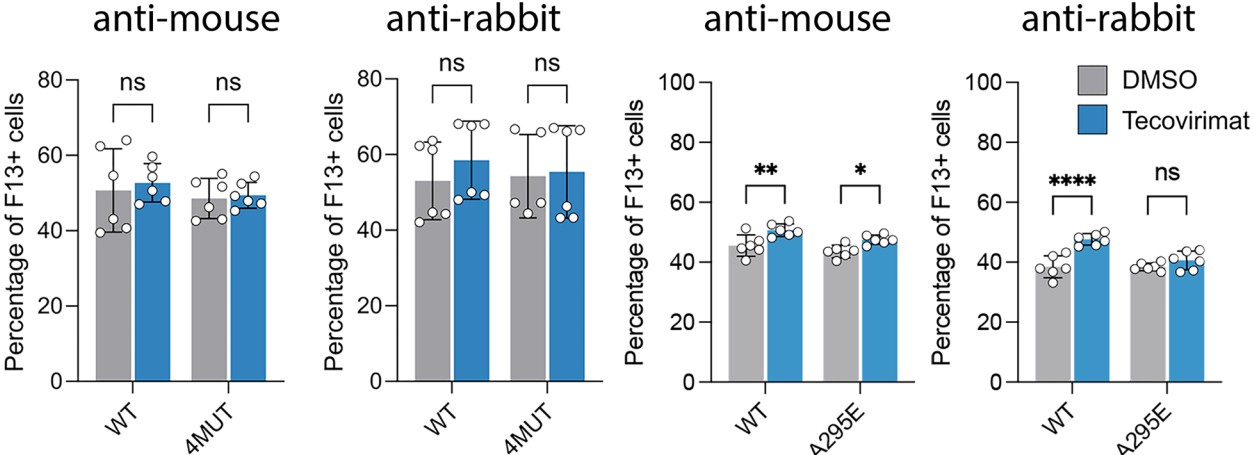

**Extended Data Fig. 7 | Image quantification methodology and percentage of F13-FLAG transfected cells. Top panels**) Quantification of the PLA signal area per cell. To measure the extent F13 dimerization, first, the number of Hoechst positive nuclei were automatically counted. Second, the PLA positive area was delimited and measured. Finally, the total PLA area was divided by the total number of nuclei. Scale bar: 50 µm. **Middle panels**) For percentage of transfection, total number of nuclei were automatically counted and delimited, then the total number of F13-FLAG positive nuclei were counted. Percentage of transfected cells was calculated by dividing the number of positive nuclei by the total number of nuclei. Scale bar: 50 µm. **Bottom panels**) Quantification of the number of F13-FLAG positive cells for the indicated treatments. Detection with anti-Mouse and anti-rabbit secondary antibodies is indicated. 7000 to 12000 cells were analysed per data point. Data are mean±sd of two independent experiments performed in triplicat (n=6). Statistical analysis: Two-Way ANOVA. ns: non-significant. *p=0.0228, **p=0.0033, ****p<0.0001.

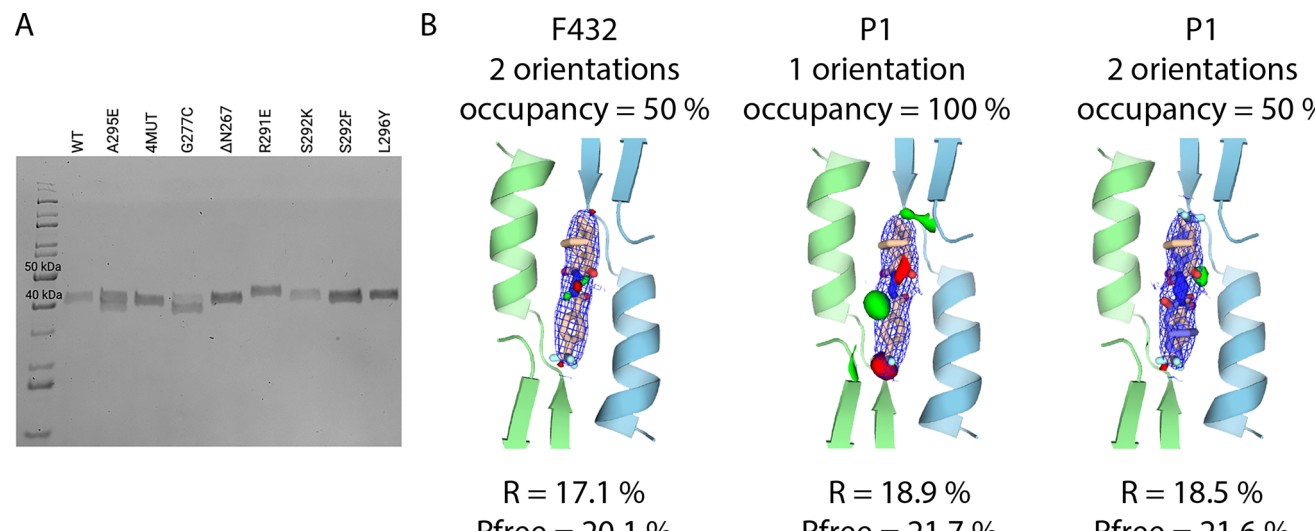

A

B

F432
2 orientations
occupancy = 50 %

R = 17.1 %
Rfree = 20.1 %

P1
1 orientation
occupancy = 100 %

R = 18.9 %
Rfree = 21.7 %

P1
2 orientations
occupancy = 50 %

R = 18.5 %
Rfree = 21.6 %

**Extended Data Fig. 8 | Proteins and crystal analysis. a)** SDS-PAGE of the proteins used in this manuscript. Molecular weight markers are shown in the left lane. In the other lanes, 1 µg of sF13 wild-type and mutants have been loaded, as indicated. For the sake of clarity, we have prepared a gel with all proteins at the same time, side by side. All of them have been analyzed at least 3 times on SDS-PAGE with identical results. **b)** Crystal structure of the sF13/tecovirimat complex processed in different space groups, as indicated. 2Fo-Fc maps contoured at 1σ are shown in blue, Fo-Fc maps contoured at +3σ and -3σ are shown in green and red, respectively. In the cubic space group (F432), a 2-fold symmetry axis passes through the center of the tecovirimat molecule, so the density represents two tecovirimat molecules with 50% occupancy each. Only one of the molecules is shown. In the central panel, we display the electron density resulting from refining the structure in P1, using a single orientation for tecovirimat (100% occupancy). Similarly, in the right panel, we process the data in P1 but refine the structure using two orientations with 50% occupancy each.

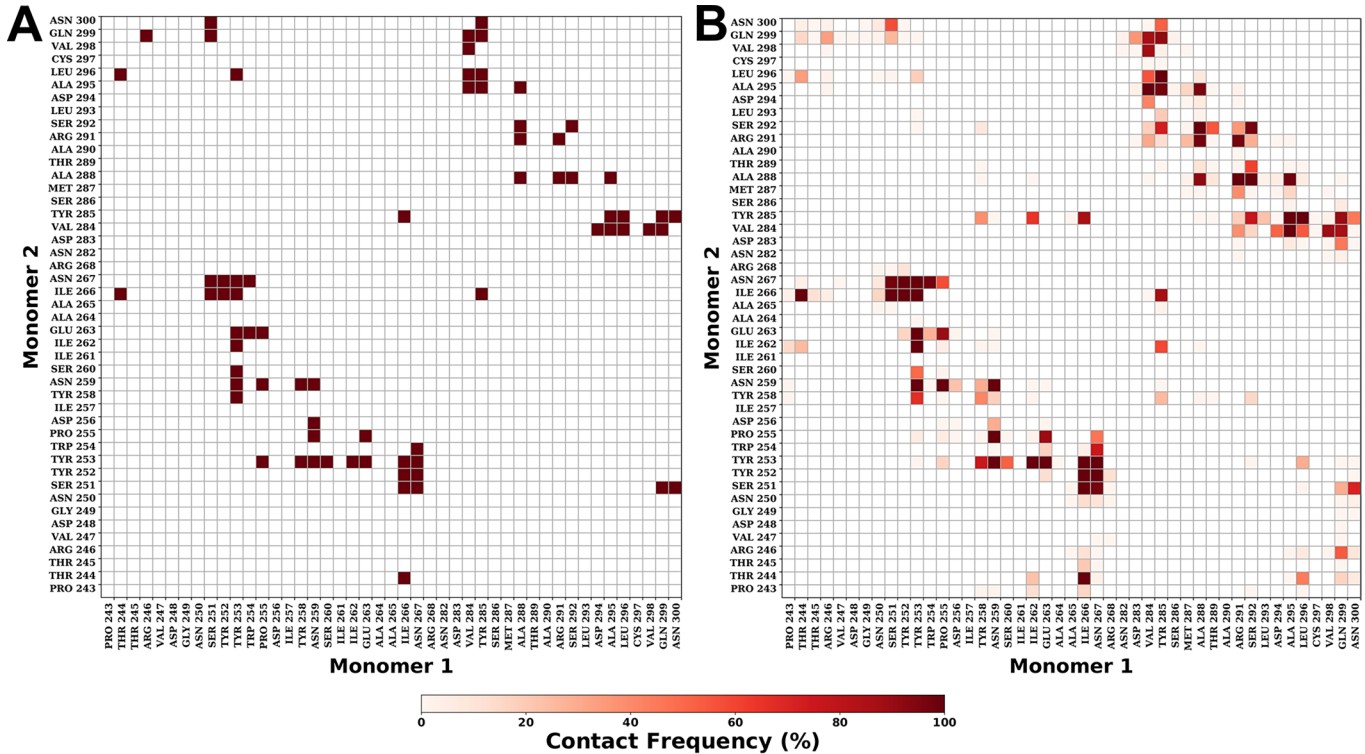

**Extended Data Fig. 9 | MD simulations of the F13 dimer on a lipid membrane.**
(**a**) Contact map showing interactions between the two monomers, calculated from the X-ray structure. (**b**) Contact map showing interactions between the two monomers, calculated from MD simulations by concatenating the last 300 ns from five repeats. The contact map highlights monomer-monomer interactions within 5 Å.

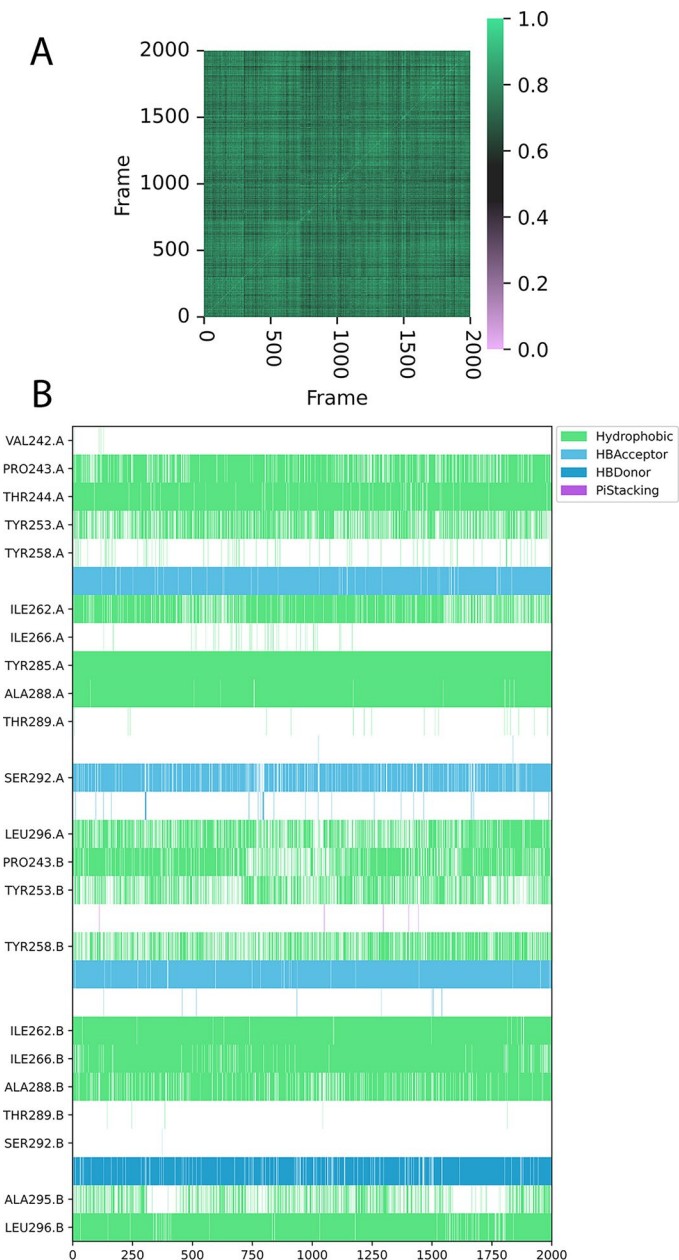

**Extended Data Fig. 10 | Protein-ligand interaction network analysis of the last 10 ns of the free simulation during the equilibration phase for pose 6-3. a**) Tanimoto similarity matrix representing ligand-protein interactions across each frame of the MD trajectory. Values range from 0 to 1, where 1 indicates the highest similarity and 0 indicates the lowest. **b**) Barcode plot of interactions. Each horizontal line represents the presence of the corresponding interaction at a specific frame of the MD trajectory.

# Reporting Summary

## Statistics

For all statistical analyses, confirm that the following items are present in the figure legend, table legend, main text, or Methods section.

| n/a | Confirmed | |
|---|---|---|
| ☐ | ☒ | The exact sample size (*n*) for each experimental group/condition, given as a discrete number and unit of measurement |
| ☐ | ☒ | A statement on whether measurements were taken from distinct samples or whether the same sample was measured repeatedly |
| ☐ | ☒ | The statistical test(s) used AND whether they are one- or two-sided *Only common tests should be described solely by name; describe more complex techniques in the Methods section.* |
| ☒ | ☐ | A description of all covariates tested |
| ☐ | ☒ | A description of any assumptions or corrections, such as tests of normality and adjustment for multiple comparisons |
| ☐ | ☒ | A full description of the statistical parameters including central tendency (e.g. means) or other basic estimates (e.g. regression coefficient) AND variation (e.g. standard deviation) or associated estimates of uncertainty (e.g. confidence intervals) |
| ☐ | ☒ | For null hypothesis testing, the test statistic (e.g. *F*, *t*, *r*) with confidence intervals, effect sizes, degrees of freedom and *P* value noted *Give P values as exact values whenever suitable.* |
| ☒ | ☐ | For Bayesian analysis, information on the choice of priors and Markov chain Monte Carlo settings |
| ☒ | ☐ | For hierarchical and complex designs, identification of the appropriate level for tests and full reporting of outcomes |
| ☐ | ☒ | Estimates of effect sizes (e.g. Cohen's *d*, Pearson's *r*), indicating how they were calculated |

*Our web collection on statistics for biologists contains articles on many of the points above.*

## Software and code

Policy information about availability of computer code

| | |
|---|---|
| Data collection | Harmony Software v4.9 (Perkin-Elmer), Acquire MP v2.3 (Refeyn), MXCuBE (version 2). All MD simulations were carried out using publicly available open source code GROMACS, versions 2021 (for dimeric F13 simulation on membrane) and 2022.4 (for F13/tecovirimat simulations). Ligand was fitted into the X-ray density using the open source tools ChimeraX 1.7.1 and Rosetta. |
| Data analysis | X-ray crystallography: XDS (version January 10, 2022), CCP4 (version 9), PHENIX (version 1.19.2-4158), PYMOL (version 3.0.3) . Mass photometry: DiscoverMP v2.3 (Refeyn). Plotting: Prism v9.0.2 (Graph Pad Software), Adobe Photoshop (v.24.5.0). Analitical ultracentrifugation: SEDNTERP (version 1.9), SEDFIT (version 16.1). Small angle x-ray scattering: ATSAS (version 3.2), GNOM (version 4.6). PLA and IF images: Signals Image Artist v1.3 (Revvity). Sequence analysis: MAFFT (version 7.505), Geneious Prime (version 2024.0.5). MD analysis: were formed using GROMACS 2021 tools and open source python tools such as alchemlyb-2.0.0 and ProLIF. TOFF 0.1.0 (https://zenodo.org/records/8189649). BindFlow has been used for absolute binding free energy calculations available at https://github.com/ale94mleon/BindFlow. A publication describing Bindflow is under preparation, while the code will be released as open source in the next months. Data were plotted using matplotlib. |

For manuscripts utilizing custom algorithms or software that are central to the research but not yet described in published literature, software must be made available to editors and reviewers. We strongly encourage code deposition in a community repository (e.g. GitHub). See the Nature Portfolio guidelines for submitting code & software for further information.

## Data

Policy information about <u>availability of data</u>

All manuscripts must include a <u>data availability statement</u>. This statement should provide the following information, where applicable:
- Accession codes, unique identifiers, or web links for publicly available datasets
- A description of any restrictions on data availability
- For clinical datasets or third party data, please ensure that the statement adheres to our <u>policy</u>

Atomic coordinates of the reported structures have been deposited in the Protein Data Bank under accession codes 9FHK, 9FHS, 9HAH, 9FJ1, 9FIZ, 9FJA, 9FJ0. All Molecular Dynamic Parameters (MDP), input topologies, coordinates, simulation control files and analysis scripts are provided. Files containing the sampled ΔH and ΔH/Δλ for ABFE are also included. All necessary files to reproduce the molecular dynamic simulations are part of the supporting information and are available at at https://zenodo.org/records/14096216. All the F13 sequences were extracted from the GISAID database at www.gisaid.org.

## Research involving human participants, their data, or biological material

Policy information about studies with <u>human participants or human data</u>. See also policy information about <u>sex, gender (identity/presentation), and sexual orientation</u> and <u>race, ethnicity and racism</u>.

| | |
|---|---|
| Reporting on sex and gender | N/A |
| Reporting on race, ethnicity, or other socially relevant groupings | N/A |
| Population characteristics | N/A |
| Recruitment | N/A |
| Ethics oversight | N/A |

Note that full information on the approval of the study protocol must also be provided in the manuscript.

# Field-specific reporting

Please select the one below that is the best fit for your research. If you are not sure, read the appropriate sections before making your selection.

☒ Life sciences ☐ Behavioural & social sciences ☐ Ecological, evolutionary & environmental sciences

For a reference copy of the document with all sections, see <u>nature.com/documents/nr-reporting-summary-flat.pdf</u>

# Life sciences study design

All studies must disclose on these points even when the disclosure is negative.

| | |
|---|---|
| Sample size | MD simulation of membrane patches: Simulation time and replication were large enough to exclude any periodic boundary artifacts or sampling issues or reproducibility issues. PLA: For each condition the experiment was performed twice in triplicates (n=6). Mass photometry: for each condition the experiment was performed in triplicates, each measurements included around 1800 to 4000 particles. Viral inhibition assay: the experiment was performed in quadruplicate. Viral plaque assay: the experiment was perfomed in duplicate and the statistics are based on multiple area measurement within each duplicate. |
| Data exclusions | No independent experiments were excluded |
| Replication | All experiments were performed and verified in multiple replicates as indicated in figure legends |
| Randomization | Sample randomization is not relevant to this study as no populations were investigated. Randomization would not have changed the results. The study is performed under controlled condition, and the reliability stems from the precision and reproducibility of the experiments rather than the use of randomization. |
| Blinding | Data collection and analysis was not performed blind because it was not needed. The study is performed under controlled condition, using purified molecules, cell lines, and viral strains. The experimental conditions were standardized and reproducible, ensuring that the outcomes were determined by the experimental variables rather than observer input. The data analysis methods, were predefined and computationally driven, eliminating the potential for investigator bias during data interpretation. |

# Reporting for specific materials, systems and methods

We require information from authors about some types of materials, experimental systems and methods used in many studies. Here, indicate whether each material, system or method listed is relevant to your study. If you are not sure if a list item applies to your research, read the appropriate section before selecting a response.

## Materials & experimental systems

| n/a | Involved in the study |
|-----|-----------------------|
| ☐ | ☒ Antibodies |
| ☐ | ☒ Eukaryotic cell lines |
| ☒ | ☐ Palaeontology and archaeology |
| ☒ | ☐ Animals and other organisms |
| ☒ | ☐ Clinical data |
| ☒ | ☐ Dual use research of concern |
| ☒ | ☐ Plants |

## Methods

| n/a | Involved in the study |
|-----|-----------------------|
| ☒ | ☐ ChIP-seq |
| ☒ | ☐ Flow cytometry |
| ☒ | ☐ MRI-based neuroimaging |

## Antibodies

| | |
|---|---|
| Antibodies used | DuoLink PLA anti-rabbit PLUS probe (cat #DUO92002, Merck) (dil = 1:5)<br>Duolink PLA anti-mouse MINUS probe (cat #DUO92004, Merck) (dil = 1:5)<br>Mouse anti-FLAG M2 (cat #F3165, Sigma-Aldrich) (dil = 1:350)<br>Rabbit anti-FLAG antibody D6W5B (cat #14793, Cell Signaling Technology) (dil = 1:500)<br>Alexa FluorTM 488 goat anti-mouse antibody (cat #A-11001, Invitrogen) (dil = 1:500)<br>Alexa FluorTM 488 goat anti-rabbit antibody (cat #A-11008, Invitrogen) (dil = 1:500) |
| Validation | Validation of all the antibodies has been performed by their providers (Merck, Sigma-Aldrich, Invitrogen and Cell Signaling Technology).<br>- DuoLink PLA anti-rabbit PLUS probe and Duolink PLA anti-mouse MINUS probe: suitable for Immunofluorescence and proximity ligation assay.<br>- Mouse anti-FLAG M2: suitable for immunoblotting, immunofluorescence, immunoprecipitation, FACS and ELISA<br>- Rabbit anti-FLAG antibody D6W5B : Suitable for western blotting, immunoprecipitation, immunoisothcemistry, immunofluorescence, flow cytometry and chropatin immunoprecipitation.<br>- Alexa FluorTM 488 goat anti-mouse antibody and Alexa FluorTM 488 goat anti-rabbit antibody: suitable for immuno istochemistry, immunocytochemistry, immunofluorescence and flow cytometry. |

## Eukaryotic cell lines

Policy information about cell lines and Sex and Gender in Research

| | |
|---|---|
| Cell line source(s) | Hela CCL2 (ATCC), BSC40 (ATCC #CRL-2761) and Vero E6 (ATCC) cells |
| Authentication | Genotyping (Eurofins) |
| Mycoplasma contamination | All cell lines were subjected to routine mycoplasma testing (Lonza™ Mycoalert™ Mycoplasma Detection Kit) and were found to be negative |
| Commonly misidentified lines<br>(See ICLAC register) | None |

## Plants

| | |
|---|---|
| Seed stocks | *Report on the source of all seed stocks or other plant material used. If applicable, state the seed stock centre and catalogue number. If plant specimens were collected from the field, describe the collection location, date and sampling procedures.* |
| Novel plant genotypes | *Describe the methods by which all novel plant genotypes were produced. This includes those generated by transgenic approaches, gene editing, chemical/radiation-based mutagenesis and hybridization. For transgenic lines, describe the transformation method, the number of independent lines analyzed and the generation upon which experiments were performed. For gene-edited lines, describe the editor used, the endogenous sequence targeted for editing, the targeting guide RNA sequence (if applicable) and how the editor was applied.* |
| Authentication | *Describe any authentication procedures for each seed stock used or novel genotype generated. Describe any experiments used to assess the effect of a mutation and, where applicable, how potential secondary effects (e.g. second site T-DNA insertions, mosiacism, off-target gene editing) were examined.* |

