## [Peer Review File · Nature Microbiology]

Structural insights into tecovirimat antiviral activity and poxvirus resistance

Corresponding Author: Dr Pablo Guardado-Calvo

Version 0:

Reviewer comments:

Reviewer #1

(Remarks to the Author)

In this paper, the authors report the crystal structure of OPXV's envelope protein F13, in both apo state and in complex with tecovirimat, a widely used drug to treat mopx. By a combination of a series of biophysical experiments, including molecular dynamics simulation, mass spectrometry, analytical ultracentrifugation, and small-angle scattering, the authors have confirmed that tecovirimat acts like a molecular glue, binding to the cavity at the interface of F13 homodimer and promoting its formation. Studies of the structure of F13 containing tecovirimat-escape mutations, as well as proximity ligation assay and producing the recombinant virus have allowed the authors to discuss the mechanism of tecovirimat resistance in mopx treatment.

This study provides molecular evidence of tecovirimat resistance in mopx treatment, with some innovative and applicable value. However, there are still some issues that the authors still need to address or improve.

1. Please show images of the crystals studied in this manuscript.
2. The figures in lines 153 and 154 are referenced incorrectly: Fig.S3A and Fig.S3B should be Fig.S4A and Fig.S4B. Also, in line 169, there is an error in the figure quotation. Please review them carefully.
3. In the structure of F13 in complex with tecovirimat and IMCBH, the authors used molecular dynamics and other methods to calculate the precise positions of tecovirimat and IMCBH. Since the positions of tecovirimat and IMCBH are critical for this study and due to the low density of these drug molecules, SAR studies are required to validate the accuracy of these calculations.
4. In Figure 3C, the EC50 value of IMCBH is 1475 nm, while in Figure 3D, the IC50 value of IMCBH is 74 nm. Considering that the difference between the corresponding values of tecovirimat is not that large, I hope the authors can discuss in more detail, to compare the binding differences between tecovirimat and IMCBH.
5. In Fig 4A, the MP experiment indicates that sF13A295E is unable to form dimers in the presence of tecovirimat. However, in Fig 4B experiment, it is shown that tecovirimat can induce sF13A295E to form dimer at high protein concentrations. These results seem puzzling, and it would be helpful if the author provided a detailed discussion on this matter.
6. The PLA signals of Flag-F13WT and Flag-F134MUT are compared in Fig. 5. What about the PLA signal of sF13A195E? I would like the authors can provide relevant data to explore whether tecovirimat can induce sF13A295E dimerization at the cellular level.
7. Please indicate the position of Q299 in Fig.4C and mark the number of alpha helices.
8. In the discussion section, lines 294 to 295, The author states "The presence of a large cavity in the dimerization interface of F13, which reduces dimer stability". It is necessary to calculate the three groups of structures in Fig.S2 before giving this conclusion.

Reviewer #2

(Remarks to the Author)

Tecovirimat is a specific orthopoxvirus inhibitor with a unique action that prevents virus spread by interacting with the F13 viral membrane protein. The current study makes a significant advance in understanding the mechanism of drug action by determining the structure of F13 alone and in complex with tecovirimat and providing evidence that the drug induces dimerization of F13. Furthermore, virus escape mutations are at the dimer interface and prevent drug-induced dimerization. The paper is written clearly so that a non-structural biologist like me was able to follow the logic – though I can't really comment on technical aspects. I feel it necessary to mention that although tecovirimat has been used to treat individuals infected with clade

IIb MPXV, a recent study showed no benefit in recovery of patients infected with pathogenic clade I virus.

Reviewer #3

(Remarks to the Author)

The study by Vernuccio et al. focuses on the structural characterization of the action of Tecovirimat. Since the mpox virus has pandemic potential, this study is important and, in my opinion, deserves to be published in Nature Microbiology. However, some statements are overstatements. For instance: "Tecovirimat, the most widely used drug to treat these infections, blocks viral egress through a poorly understood mechanism." In fact, the mechanism is known (as detailed in the section "Mechanism of Action" in PMC9765296). Although I agree that the atomic details of the mechanism of action were not known.

Nevertheless, I was asked to primary focus on the crystallography. Which is well done and I have only one very few comments.

1) "However, because the pocket is crossed by a crystallographic two-fold axis and tecovirimat is an asymmetric molecule, the resulting electron density is featureless, preventing the accurate modeling of the molecule using only the crystallographic density."

No need to blame crystallographic symmetry. You can process the data in the P1 group with two molecules in the asymmetric unit and the crystallographic 2-fold axis symmetry will disappear and you will get a two-fold non-crystallographic symmetry axis to construct the F13 dimer. But the problem will be the same.

Since the dimer is symmetrical and tecovirimat is not, the observed map is an average of the two binding poses that tecovirimat must exhibit. That obviously makes fitting tecovirimat in the density more difficult. You can fit tecovirimat in two poses that will be symmetry related with 50% occupancy. However, the current approach is also fine. I would suggest to try this approach and present the results as an SI Figure. It will be interesting to see if the Rfactors change (drop).

Also the density for tecovirimat is 2-fold averaged, it is features rather due mediocre resolution (I guess 2.9Å for $l/\sigma = 2$). At a high resolution the details would be well visible despite the 2-fold averaging.

2) The crystallographic Table is well done and detailed. Weak reflections were included which is the current standard as long as $CC1/2$ is $\geq \sim 0.5$. I only miss the resolution where $l/\sigma = 2$ which is what is historically called the resolution.

3) Figure 3 would be nicer if a SAXS-based envelope would be calculated and how the crystallographic dimer fits the envelope would be shown.

Decision Letter:

11th October 2024

Dear Pablo,

Thank you for your patience while your manuscript "MECHANISMS OF TECOVIRIMAT ANTIVIRAL ACTIVITY AND POXVIRUS RESISTANCE" was under peer-review at Nature Microbiology. It has now been seen by 3 referees, whose expertise and comments you will find at the end of this email. Although they find your work of some potential interest, they have raised a number of concerns that will need to be addressed before we can consider publication of the work in Nature Microbiology.

In particular, referee #1 asks to perform SAR studies to validate your findings and to include sF13A195E in your PLA comparison. Please also discuss the differences in Tecovirimat effectivity against Clade I and II MPXVs in humans, as raised by referee #2. The other referees comments should be straightforward to address.

Should further experimental data allow you to address these criticisms, we would be happy to look at a revised manuscript.

Please include a data availability statement as a separate section after Methods but before references, under the heading "Data Availability". This section should inform readers about the availability of the data used to support the conclusions of your study. This information includes accession codes to public repositories (data banks for protein, DNA or RNA sequences, microarray, proteomics data etc...), references to source data published alongside the paper, unique identifiers such as URLs to data

repository entries, or data set DOIs, and any other statement about data availability. At a minimum, you should include the following statement: "The data that support the findings of this study are available from the corresponding author upon request", mentioning any restrictions on availability. If DOIs are provided, we also strongly encourage including these in the Reference list (authors, title, publisher (repository name), identifier, year). For more guidance on how to write this section please see: <http://www.nature.com/authors/policies/data/data-availability-statements-data-citations.pdf>

* If you have not done so already we suggest that you begin to revise your manuscript so that it conforms to our Article format instructions at <http://www.nature.com/nmicrobiol/info/final-submission>. Refer also to any guidelines provided in this letter.

When submitting the revised version of your manuscript, please pay close attention to our [href="https://www.nature.com/nature-portfolio/editorial-policies/image-integrity">Digital Image Integrity Guidelines.](https://www.nature.com/nature-portfolio/editorial-policies/image-integrity) and to the following points below:

Link Redacted

Note: This url links to your confidential homepage and associated information about manuscripts you may have submitted or be reviewing for us. If you wish to forward this e-mail to co-authors, please delete this link to your homepage first.

Nature Microbiology is committed to improving transparency in authorship. As part of our efforts in this direction, we are now requesting that all authors identified as 'corresponding author' on published papers create and link their Open Researcher and Contributor Identifier (ORCID) with their account on the Manuscript Tracking System (MTS), prior to acceptance. This applies to primary research papers only. ORCID helps the scientific community achieve unambiguous attribution of all scholarly contributions. You can create and link your ORCID from the home page of the MTS by clicking on 'Modify my Springer Nature account'. For more information please visit [please visit www.springernature.com/orcid](http://www.springernature.com/orcid).

If you wish to submit a suitably revised manuscript we would hope to receive it within 3 months. If you cannot send it within this time, please let us know the expected timeline. We will be happy to consider your revision, even if a similar study has been accepted for publication at Nature Microbiology or published elsewhere (up to a maximum of 3 months).

Yours sincerely,

Reviewer Expertise:

Referee #1: Structural biology, CryoEM, drug design

Referee #2: Poxviruses, HPI

Referee #3: Structural biology, X-ray, antivirals

Reviewer Comments:

Reviewer #1 (Remarks to the Author):

In this paper, the authors report the crystal structure of OPXV's envelope protein F13, in both apo state and in complex with tecovirimat, a widely used drug to treat mpox. By a combination of a series of biophysical experiments, including molecular dynamics simulation, mass spectrometry, analytical ultracentrifugation, and small-angle scattering, the authors have confirmed that tecovirimat acts like a molecular glue, binding to the cavity at the interface of F13 homodimer and promoting its formation. Studies of the structure of F13 containing tecovirimat-escape mutations, as well as proximity ligation assay and producing the

recombinant virus have allowed the authors to discuss the mechanism of tecovirimat resistance in mopx treatment.

This study provides molecular evidence of tecovirimat resistance in mopx treatment, with some innovative and applicable value. However, there are still some issues that the authors still need to address or improve.

1. Please show images of the crystals studied in this manuscript.
2. The figures in lines 153 and 154 are referenced incorrectly: Fig.S3A and Fig.S3B should be Fig.S4A and Fig.S4B. Also, in line 169, there is an error in the figure quotation. Please review them carefully.
3. In the structure of F13 in complex with tecovirimat and IMCBH, the authors used molecular dynamics and other methods to calculate the precise positions of tecovirimat and IMCBH. Since the positions of tecovirimat and IMCBH are critical for this study and due to the low density of these drug molecules, SAR studies are required to validate the accuracy of these calculations.
4. In Figure 3C, the EC50 value of IMCBH is 1475 nm, while in Figure 3D, the IC50 value of IMCBH is 74 nm. Considering that the difference between the corresponding values of tecovirimat is not that large, I hope the authors can discuss in more detail, to compare the binding differences between tecovirimat and IMCBH.
5. In Fig 4A, the MP experiment indicates that sF13A295E is unable to form dimers in the presence of tecovirimat. However, in Fig 4B experiment, it is shown that tecovirimat can induce sF13A295E to form dimer at high protein concentrations. These results seem puzzling, and it would be helpful if the author provided a detailed discussion on this matter.
6. The PLA signals of Flag-F13WT and Flag-F134MUT are compared in Fig. 5. What about the PLA signal of sF13A195E? I would like the authors can provide relevant data to explore whether tecovirimat can induce sF13A295E dimerization at the cellular level.
7. Please indicate the position of Q299 in Fig.4C and mark the number of alpha helices.
8. In the discussion section, lines 294 to 295, The author states "The presence of a large cavity in the dimerization interface of F13, which reduces dimer stability". It is necessary to calculate the three groups of structures in Fig.S2 before giving this conclusion.

Reviewer #2 (Remarks to the Author):

Tecovirimat is a specific orthopoxvirus inhibitor with a unique action that prevents virus spread by interacting with the F13 viral membrane protein. The current study makes a significant advance in understanding the mechanism of drug action by determining the structure of F13 alone and in complex with tecovirimat and providing evidence that the drug induces dimerization of F13. Furthermore, virus escape mutations are at the dimer interface and prevent drug-induced dimerization. The paper is written clearly so that a non-structural biologist like me was able to follow the logic – though I can't really comment on technical aspects. I feel it necessary to mention that although tecovirimat has been used to treat individuals infected with clade IIb MPXV, a recent study showed no benefit in recovery of patients infected with pathogenic clade I virus.

Reviewer #3 (Remarks to the Author):

The study by Vernuccio et al. focuses on the structural characterization of the action of Tecovirimat. Since the mpox virus has pandemic potential, this study is important and, in my opinion, deserves to be published in Nature Microbiology. However, some statements are overstatements. For instance: "Tecovirimat, the most widely used drug to treat these infections, blocks viral egress through a poorly understood mechanism." In fact, the mechanism is known (as detailed in the section "Mechanism of Action" in PMC9765296). Although I agree that the atomic details of the mechanism of action were not known.

Nevertheless, I was asked to primary focus on the crystallography. Which is well done and I have only one very few comments.

1) "However, because the pocket is crossed by a crystallographic two-fold axis and tecovirimat is an asymmetric molecule, the resulting electron density is featureless, preventing the accurate modeling of the molecule using only the crystallographic density."

No need to blame crystallographic symmetry. You can process the data in the P1 group with two molecules in the asymmetric unit and the crystallographic 2-fold axis symmetry will disappear and you will get a two-fold non-crystallographic symmetry axis to construct the F13 dimer. But the problem will be the same.

Since the dimer is symmetrical and tecovirimat is not, the observed map is an average of the two binding poses that tecovirimat must exhibit. That obviously makes fitting tecovirimat in the density more difficult. You can fit tecovirimat in two poses that will be symmetry related with 50% occupancy. However, the current approach is also fine. I would suggest to try this approach and present the results as an SI Figure. It will be interesting to see if the Rfactors change (drop).

Also the density for tecovirimat is 2-fold averaged, it is features rather due mediocre resolution (I guess 2.9Å for $1/\sigma = 2$). At

a high resolution the details would be well visible despite the 2-fold averaging.

2) The crystallographic Table is well done and detailed. Weak reflections were included which is the current standard as long as $CC1/2$ is $\geq \sim 0.5$. I only miss the resolution where $l/\sigma = 2$ which is what is historically called the resolution.

3) Figure 3 would be nicer if a SAXS-based envelope would be calculated and how the crystallographic dimer fits the envelope would be shown.

Version 1:

Reviewer comments:

Reviewer #1

(Remarks to the Author)

Thanks for your revision. I am pleased with it.

Reviewer #2

(Remarks to the Author)

No further comments

Reviewer #3

(Remarks to the Author)

The crystallographic part of the manuscript, which I was specifically asked to pay attention to, is now perfect. The authors addressed all my comments.

Decision Letter:

Our ref: NMICROBIOL-24082685A

11th December 2024

Dear Pablo,

Thank you for submitting your revised manuscript "MECHANISMS OF TECOVIRIMAT ANTIVIRAL ACTIVITY AND POXVIRUS RESISTANCE" (NMICROBIOL-24082685A). It has now been seen by the original referees and their comments are below. The reviewers find that the paper has improved in revision, and therefore we'll be happy in principle to publish it in Nature Microbiology, pending minor revisions to comply with our editorial and formatting guidelines.

Thank you again for your interest in Nature Microbiology Please do not hesitate to contact me if you have any questions.

Sincerely,

Reviewer #1 (Remarks to the Author):

Thanks for your revision. I am pleased with it.

Reviewer #2 (Remarks to the Author):

No further comments

Reviewer #3 (Remarks to the Author):

The crystallographic part of the manuscript, which I was specifically asked to pay attention to, is now perfect. The authors addressed all my comments.

Version 2:

Decision Letter:

14th January 2025

Dear Pablo,

I am pleased to accept your Article "Structural insights into tecovirimat antiviral activity and poxvirus resistance" for publication in Nature Microbiology. Thank you for having chosen to submit your work to us and many congratulations.

Authors may need to take specific actions to achieve [compliance](https://www.springernature.com/gp/open-research/funding/policy-compliance-faqs) with funder and institutional open access mandates. If your research is supported by a funder that requires immediate open access (e.g. according to [Plan S principles](https://www.springernature.com/gp/open-research/plan-s-compliance)) then you should select the gold OA route, and we will direct you to the compliant route where possible. For authors selecting the subscription publication route, the journal's standard licensing terms will need to be accepted, including [self-archiving policies](https://www.nature.com/nature-portfolio/editorial-policies/self-archiving-and-license-to-publish). Those licensing terms will supersede any other terms that the author or any third party may assert apply to any version of the manuscript.

As soon as your article is published, you will receive an automated email with your shareable link. Congrats again to you and your co-authors! I am looking forward to seeing your paper published.

With kind regards,

P.S. Click on the following link if you would like to recommend Nature Microbiology to your librarian
<http://www.nature.com/subscriptions/recommend.html#forms>

** Visit the Springer Nature Editorial and Publishing website at http://editorial-jobs.springernature.com?utm_source=ejP_NMicro_email&utm_medium=ejP_NMicro_email&utm_campaign=ejp_NMicro for more information about our career opportunities. If you have any questions please click [here](mailto:editorial.publishing.jobs@springernature.com).

Open Access This Peer Review File is licensed under a Creative Commons Attribution 4.0 International License, which permits use, sharing, adaptation, distribution and reproduction in any medium or format, as long as you give appropriate credit to the original author(s) and the source, provide a link to the Creative Commons license, and indicate if changes were made. In cases where reviewers are anonymous, credit should be given to 'Anonymous Referee' and the source. The images or other third party material in this Peer Review File are included in the article's Creative Commons license, unless indicated otherwise in a credit line to the material. If material is not included in the article's Creative Commons license and your intended use is not permitted by statutory regulation or exceeds the permitted use, you will need to obtain permission directly from the copyright holder.

Reviewer #1:

In this paper, the authors report the crystal structure of OPXV's envelope protein F13, in both apo state and in complex with tecovirimat, a widely used drug to treat mpx. By a combination of a series of biophysical experiments, including molecular dynamics simulation, mass spectrometry, analytical ultracentrifugation, and small-angle scattering, the authors have confirmed that tecovirimat acts like a molecular glue, binding to the cavity at the interface of F13 homodimer and promoting its formation. Studies of the structure of F13 containing tecovirimat-escape mutations, as well as proximity ligation assay and producing the recombinant virus have allowed the authors to discuss the mechanism of tecovirimat resistance in mpx treatment.

This study provides molecular evidence of tecovirimat resistance in mpx treatment, with some innovative and applicable value. However, there are still some issues that the authors still need to address or improve.

1. Please show images of the crystals studied in this manuscript..

We have included a panel A in Figure 2 showing a picture of the cubic crystals used in this manuscript.

Fig. 2. Tecovirimat binding site. (A) Cubic crystals used to obtain the structure of the sF13/tecovirimat and sF13/IMCBH complexes.

2. The figures in lines 153 and 154 are referenced incorrectly: Fig.S3A and Fig.S3B should be Fig.S4A and Fig.S4B. Also, in line 169, there is an error in the figure quotation. Please review them carefully.

We thank the reviewer for finding these errors. We have corrected them and checked that there are no other in the text.

3. In the structure of F13 in complex with tecovirimat and IMCBH, the authors used molecular dynamics and other methods to calculate the precise positions of tecovirimat and IMCBH. Since the positions of tecovirimat and IMCBH are critical for this study and due to the low density of these drug molecules, SAR studies are required to validate the accuracy of these calculations.

We thank the reviewer for the suggestion. We agree that validating the tecovirimat binding pose would strengthen our article. To address this, we have performed the following SAR study. First, we have selected (from ref. 24) seven structurally similar derivatives of tecovirimat with reported EC50 values. Then, we have aligned these compounds to the tecovirimat binding pose, and calculate the free binding energy values (ΔG_{calc}). In parallel, we estimated experimental free binding energy values (ΔG_{exp}) from the reported EC50 values (see Methods for details). As shown in Fig. S4 (copied below), the calculated and experimental ΔG values have good agreement, which validates the tecovirimat binding pose and the FEP protocol. This result also suggests that these structurally similar derivatives adopt the same pose.

We discuss these results in lines 134-141:

“To validate our approach, we evaluated a set of seven structurally similar ligands with published EC50 values alongside tecovirimat^{23,24}. We aligned each ligand to the tecovirimat pose proposed herein and performed ABFE calculations. After adjusting a constant offset that accounts for the free energy cost of dimerization and a putative proportionality between inhibitory concentration and ligand activity (see Methods), we obtained good agreement between the computed ABFE values and experimental data (Fig. S4), as quantified by Pearson and Spearman correlation coefficients of 0.82 and 0.74, respectively.”

We have also added an additional supplementary figure (S4) where we show the correlation between calculated and experimental values as well as the structure of the different compounds analyzed.

Figure S4. Validation of the tecovirimat conformation using seven structurally similar ligands. Eight ligands, including tecovirimat, were aligned to the proposed tecovirimat binding pose, and the free energy of protein dimerization in the presence of each ligand was calculated (see Methods). Pearson, Kendall, and Spearman correlation coefficients are reported alongside root mean square error (RMSE), mean signed error (MSE), and mean unsigned error (MUE). Ligands are identified in the validation plot by numbers taken from²⁴. Top right: 3D alignment of the eight molecules within the binding pocket. Bottom: chemical structures of the ligands with the corresponding EC_{50}^{VACV} values. The alignment was rendered with PyMOL⁸⁸ and chemical structures were drawn with RDKit⁸⁹.

Figure S4. Validation of the tecovirimat conformation using seven structurally similar ligands. Eight ligands, including tecovirimat, were aligned to the proposed tecovirimat binding pose, and the free energy of protein dimerization in the presence of each ligand was calculated (see Methods). Pearson, Kendall, and Spearman correlation coefficients are reported alongside root mean square error (RMSE), mean signed error (MSE), and mean unsigned error (MUE). Ligands are identified in the validation plot by numbers taken from²⁴. Top right: 3D alignment of the eight molecules within the binding pocket. Bottom: chemical structures of the ligands with the corresponding EC_{50}^{VACV} values. The alignment was rendered with PyMOL⁸⁸ and chemical structures were drawn with RDKit⁸⁹.

4. In Figure 3C, the EC₅₀ value of IMCBH is 1475 nM, while in Figure 3D, the IC₅₀ value of IMCBH is 74 nM. Considering that the difference between the corresponding values of tecovirimat is not that large, I hope the authors can discuss in more detail, to compare the binding differences between tecovirimat and IMCBH.

We appreciate the reviewer's comment and agree that this discrepancy deserves a comment. IMCBH and tecovirimat are quite different and it is difficult to have a good correlation between their activity in solution and their antiviral activity. The activity in solution depends mostly on their affinity for sF13 but the antiviral activity is influenced also by additional factors, such as their ability to cross the cell membrane or their intracellular distribution. Since the molecules differ so much, these factors can vary considerably.

To discuss this, we have included a paragraph in the text (lines 187-190)

“Tecovirimat is approximately 15 times more active in solution than IMCBH, but its antiviral activity is only 5 times better. This discrepancy could be explained by factors unrelated to their biochemical activities, such as differences in membrane permeability or cellular distribution.”

5. In Fig 4A, the MP experiment indicates that sF13A295E is unable to form dimers in the presence of tecovirimat. However, in Fig 4B experiment, it is shown that tecovirimat can induce sF13A295E to form dimer at high protein concentrations. These results seem puzzling, and it would be helpful if the author provided a detailed discussion on this matter.

The differences between the MP (mass-photometry) and AUC (analytical ultracentrifugation) experiments comes from the difference in sF13 concentration used in both experiments: 25 nM in MP and 10 μM in AUC. We used both approaches because in the initial MP experiments we did not observe difference in activity between sF13^{A295E} and sF13^{4MUT} mutants, despite we know 4MUT escapes tecovirimat activity more effectively than A295E. We hypothesised that the problem is that the sF13 concentration is too low to detect sF13^{A295E} dimerization but we cannot increase the protein concentration in the MP experiment. To address this limitation, we performed AUC experiments with a 400-fold higher sF13 concentration. These new experiments revealed that tecovirimat induces partial dimerization of sF13^{A295E} but not of sF13^{4MUT}, consistent with the reported ΔEC₅₀ values.

We have rewritten a paragraph (lines 216-222) to explain all this better.

“To complement the MP data, we repeated the AUC experiments reported above using sF13^{A295E} and sF13^{4MUT} (Fig. 4B and Fig. S6). In contrast to the MP experiments, which were performed using a dilute solution of sF13 (25 nM), the AUC experiments were performed at a higher concentration (10 μM), which facilitates protein homodimerization, to detect weaker tecovirimat activities. Thus, in the AUC experiments, we observed that the drug induced partial dimerization of sF13A295E, but not of sF134MUT, which aligns with the ΔEC₅₀ values reported for A295E and 4MUT (Table S3).“

6. The PLA signals of Flag-F13WT and Flag-F134MUT are compared in Fig. 5. What about the PLA signal of sF13A195E? I would like the authors can provide relevant data to explore whether tecovirimat can induce sF13A295E dimerization at the cellular level.

As the reviewer has suggested, we have performed PLA experiments using the A295E mutant. What we have observed is that tecovirimat induces a slight dimerisation of this mutant, which contrasts with the intense dimerisation of the wild-type protein and the lack of dimerisation of the 4MUT mutant. This result is completely in line with the results obtained in solution and with the structure of the A295E mutant, which shows that A295E can still bind tecovirimat, although less efficiently. This result is also in line with the ΔEC_{50} values reported for A295E (110-190) and 4MUT (>29000). We have updated the text to discuss these results and figure 5.

Lines 251-254: “In line with the AUC experiments, tecovirimat induced a strong increase in the PLA signal in cells expressing Flag-F13^{WT}, a slight increase in cells expressing Flag-F13^{A295E} and none in cells expressing Flag-F13^{4MUT} (Fig. 5B and S7).”

Fig. 5

7. Please indicate the position of Q299 in Fig.4C and mark the number of alpha helices.

We have modified the figure to indicate the position of Q299 in the Fig. 4C and labelled the alpha helices.

8. In the discussion section, lines 294 to 295, The author states "The presence of a large cavity in the dimerization interface of F13, which reduces dimer stability". It is necessary to calculate the three groups of structures in Fig.S2 before giving this conclusion.

We thank the reviewer for this comment. Crystal structures of PLD3 and PLD4 dimers have been previously reported (refs. 19 and 34), and the PLD3 dimer has been characterized in solution using SEC and AUC (ref. 34). We have re-written a paragraph in the discussion (lines 311-315) to clarify this information.

“Despite this evolutionary link, there is a significant difference between the homodimer of F13 and its eukaryotic counterparts. The active form of PLD3 is a stable dimer in solution³⁴, but the soluble version of F13 is mostly monomeric. Structural analysis revealed a large cavity at F13's dimerization interface, which is absent in PLD3 and PLD4 homodimers. We hypothesized that this cavity reduces the stability of the F13 dimer.”

Reviewer #2:

Tecovirimat is a specific orthopoxvirus inhibitor with a unique action that prevents virus spread by interacting with the F13 viral membrane protein. The current study makes a significant advance in understanding the mechanism of drug action by determining the structure of F13 alone and in complex with tecovirimat and providing evidence that the drug induces dimerization of F13. Furthermore, virus escape mutations are at the dimer interface and prevent drug-induced dimerization. The paper is written clearly so that a non-structural biologist like me was able to follow the logic – though I can't really comment on technical aspects. I feel it necessary to mention that although tecovirimat has been used to treat individuals infected with clade IIb MPXV, a recent study showed no benefit in recovery of patients infected with pathogenic clade I virus.

We thank the reviewer for the comments and agree that it is necessary to mention the results of the clinical trial showing that tecovirimat does not accelerate the recovery of patients infected with clade I MPXV. For this, we have introduced two modifications in the text. The first in the introduction (lines 61-63) to highlight that tecovirimat has been used to treat patients infected with clade IIb strains.

“Tecovirimat is an inhibitor of MV wrapping and has been widely used to treat mpox patients infected with clade IIb strains during the 2022 outbreak.”

The second is in the discussion (lines 348-356) to discuss the results of the clinical trial.

“While we were preparing this paper, early results from the PALM 007 clinical trial were released³⁵, showing that tecovirimat did not accelerate recovery in patients infected with MPXV clade I. Based on the structure of the sF13/tecovirimat complex, we did not identify any mutations in clade I strains that could explain this lack of activity, and it is unclear if these findings apply to clade II strains. Additional clinical trials will examine tecovirimat's effectiveness on different strains and when administered early in infection. However, these results emphasize the need for new antiviral drugs. The structural data, MD simulations, and the battery of assays developed here will pave the way for the development of these novel antivirals active against tecovirimat-resistant strains.”

Reviewer #3:

The study by Vernuccio et al. focuses on the structural characterization of the action of Tecovirimat. Since the mpox virus has pandemic potential, this study is important and, in my opinion, deserves to be published in Nature Microbiology. However, some statements are overstatements. For instance: “Tecovirimat, the most widely used drug to treat these infections, blocks viral egress through a poorly understood mechanism.” In fact, the mechanism is known (as detailed in the section "Mechanism of Action" in PMC9765296). Although I agree that the atomic details of the mechanism of action were not known.

We thank the reviewer for the kind words and the support of our work. The reviewer is right, the mechanism of action of tecovirimat is known, what were unknown are the structural bases that mediate it. We have reworded the abstract to be more explicit and to avoid overstatements.

“Tecovirimat, the most widely used drug to treat mpox, blocks viral egress by targeting the viral phospholipase F13, although the structural details are unknown. Mutations in the F13 gene can result in resistance to tecovirimat, raising public health concerns.”

Nevertheless, I was asked to primary focus on the crystallography. Which is well done and I have only one very few comments.

1) “However, because the pocket is crossed by a crystallographic two-fold axis and tecovirimat is an asymmetric molecule, the resulting electron density is featureless, preventing the accurate modeling of the molecule using only the crystallographic density.”

No need to blame crystallographic symmetry. You can process the data in the P1 group with two molecules in the asymmetric unit and the crystallographic 2-fold axis symmetry will disappear and you will get a two-fold non-crystallographic symmetry axis to construct the F13 dimer. But the problem will be the same.

Since the dimer is symmetrical and tecovirimat is not, the observed map is an average of the two binding poses that tecovirimat must exhibit. That obviously makes fitting tecovirimat in the density more difficult. You can fit tecovirimat in two poses that will be symmetry related with 50% occupancy. However, the current approach is also fine. I would suggest to try this approach and present the results as an SI Figure. It will be interesting to see if the Rfactors change (drop).

Also the density for tecovirimat is 2-fold averaged, it is features rather due mediocre resolution (I guess 2.9Å for $1/\sigma = 2$). At a high resolution the details would be well visible despite the 2-fold averaging.

Again we thank the reviewer for his comments, which we have found very useful. In response:

- a) we have modified the text not to blame crystallographic symmetry, but the fact that tecovirimat can bind the pocket using (at least) two symmetric poses (lines 116-119):

“However, because the pocket is symmetrical and tecovirimat is an asymmetric molecule, the resulting electron density, an average of two symmetrical poses, is featureless, preventing accurate modeling of the molecule using only the crystallographic density.”

- b) We have carefully re-processed the data using the latest version of XDS and have obtained a (marginal) gain in resolution from 2.7 to 2.6 Å. We have re-refined the structure using this data. The density for the tecovirimat remains featureless. We have updated the crystallographic table.
- c) We have reprocessed the dataset in P1 to analyze the density in the absence of axis 2. The data were refined in two distinct ways: (1) with tecovirimat modeled in a single orientation with 100% occupancy, and (2) with tecovirimat modeled in two orientations, each at 50% occupancy, as in the cubic space group. We observed that refinement with two tecovirimat molecules rotated 180 degrees yielded better R-factors and reduced residual densities. A detailed explanation of this procedure is now included in the Methods section (lines 467–478), along with a new Figure S8 (see below).

“The electron density for tecovirimat does not show clear features. We hypothesize that this is because the binding pocket is symmetric, while the molecule itself is not, allowing it to adopt two indistinguishable orientations rotated by 180 degrees. However, it is also possible that this is a crystallographic artifact, with tecovirimat only entering in a single orientation and the density is featureless because of the presence of a 2-fold symmetry axis crossing the molecule. To investigate this, we reprocessed the cubic crystals in the space group P1 and refined the tecovirimat molecule in two different ways: in a single orientation with 100% occupancy, and in two orientations rotated by 180 degrees with 50% occupancy each, mimicking what is observed in the cubic space group. When comparing the two refinements (Fig. S8B), we observed improved R-factors and reduced residual densities when using the model with two rotated molecules, supporting the hypothesis”

Fig. S8. B) Crystal structure of the sF13/tecovirimat complex processed in different space groups, as indicated. 2Fo-Fc maps contoured at 1σ are shown in blue, Fo-Fc maps

contoured at $+3\sigma$ and -3σ are shown in green and red, respectively. In the cubic space group ($F432$), a 2-fold symmetry axis passes through the center of the tecovirimat molecule, so the density represents two tecovirimat molecules with 50% occupancy each. Only one of the molecules is shown. In the central panel, we display the electron density resulting from refining the structure in $P1$, using a single orientation for tecovirimat (100% occupancy). Similarly, in the right panel, we process the data in $P1$ but refine the structure using two orientations with 50% occupancy each.

2) The crystallographic Table is well done and detailed. Weak reflections were included which is the current standard as long as $CC1/2$ is $\geq \sim 0.5$. I only miss the resolution where $I/\sigma = 2$ which is what is historically called the resolution.

We have updated table S1 to include the estimated resolution from $I/\sigma = 2$ for all structures included in this manuscript.

3) Figure 3 would be nicer if a SAXS-based envelope would be calculated and how the crystallographic dimer fits the envelope would be shown.

We fully agree with the suggestion. We have now included a new panel in figure 3 where we show the atom dummy derived from the SAXS data, as well as the properly oriented sF13 dimer model.

We have modified the text:

Lines 166-170: "For further structural insight, we compared the experimental SAXS curve with the theoretical curves of the F13 monomer and dimer (Fig. 3B) and the *ab initio* model derived from the SAXS curve with the crystallographic F13 dimer (Fig. 3C). We found in both analyses that the crystallographic dimer fitted well to the experimental SAXS curve."

- Please list your supplementary tables and reporting summary in your Inventory.

Done

- We can only allow a maximum of 10 extended data figures. One of your extended data figures will need to be added as supplementary information. Please also ensure that your inventory and manuscript is updated to reflect this.

We have transformed Extended data 11 to Supplementary Figure 1. We have included this figure in the Supplementary Information file. We have updated the manuscript.

- Please make Table S8 an excel sheet.

We have included all the supplementary Tables as PDFs in the Supplementary Information file.

• Please refrain from using words such as new/novel/first, when referring to the scientific findings. See line 271.

We have re-written the sentence in this way:

Here, we present a structural study of viral phospholipase F13, including its interaction with tecovirimat.

• Please briefly describe the origin of the wild-type VACV-WR25, rVACV mutants and MPXV clade IIb viruses (where you received them from, not only citation). Please also describe the biosafety measurements you were adhering to when working with the MPXV clade IIb isolate and whether this was approved by the Institutional board.

We have modified the text to include the origin of all viruses used and the biosafety measures used with MPXV as follows.

Confluent BSC40 cells were infected with wild-type VACV-WR25 (provided by Jason Mercer (University of Birmingham)) or rVACV mutants generated from a modified VACV-WR (vNotI/tk) strain (originally provided by B. Moss to K. Chandran, who further modified it by incorporating an mCherry reporter into the tk locus) at a 10-fold dilution in DMEM with 2.5% FBS for 1 hour at 37°C.

The experiments with MPXV were done using the clade IIb strain MPXV/2022/FR/CMIP, which was isolated at the Institut Pasteur (France) in 2022. All experiments were conducted under strict BSL3 conditions according to the French regulations on dual use pathogens.